# Long noncoding RNA *PAHAL* modulates locust behavioural plasticity through the feedback regulation of dopamine biosynthesis

**Xia Zhang**[1,2,3], **Ya'nan Xu**[1], **Bing Chen**[1,4]*, **Le Kang**[1,2,3,4]*

**1** State Key Laboratory of Integrated Management of Pest Insects and Rodents, Institute of Zoology, Chinese Academy of Sciences, Beijing, China, **2** Beijing Institute of Life Sciences, Chinese Academy of Sciences, Beijing, China, **3** CAS Center for Excellence in Biotic Interactions, University of Chinese Academy of Sciences, Beijing, China, **4** College of Life Sciences, Hebei University, Baoding, China

* chenbing@hbu.edu.cn (BC); lkang@ioz.ac.cn (KL)

## Abstract

Some long noncoding RNAs (lncRNAs) are specifically expressed in brain cells, implying their neural and behavioural functions. However, how lncRNAs contribute to neural regulatory networks governing the precise behaviour of animals is less explored. Here, we report the regulatory mechanism of the nuclear-enriched lncRNA *PAHAL* for dopamine biosynthesis and behavioural adjustment in migratory locusts (*Locusta migratoria*), a species with extreme behavioral plasticity. *PAHAL* is transcribed from the sense (coding) strand of the gene encoding phenylalanine hydroxylase (*PAH*), which is responsible for the synthesis of dopamine from phenylalanine. *PAHAL* positively regulates *PAH* expression resulting in dopamine production in the brain. In addition, *PAHAL* modulates locust behavioral aggregation in a population density-dependent manner. Mechanistically, *PAHAL* mediates *PAH* transcriptional activation by recruiting serine/arginine-rich splicing factor 2 (SRSF2), a transcription/splicing factor, to the *PAH* proximal promoter. The co-activation effect of *PAHAL* requires the interaction of the *PAHAL*/SRSF2 complex with the promoter-associated nascent RNA of *PAH*. Thus, the data support a model of feedback modulation of animal behavioural plasticity by an lncRNA. In this model, the lncRNA mediates neurotransmitter metabolism through orchestrating a local transcriptional loop.

## Author summary

The neurotransmitter dopamine is crucial for the neuronal and behavioral response in animals. Phenylalanine hydroxylase (PAH) is involved in dopamine biosynthesis and behavioral regulation in the migratory locust. However, the molecular mechanism for the fine tuning of *PAH* expression in behavioral response remains ambiguous. Here we discovered a nuclear-enriched lncRNA *PAHAL* that is transcribed from the coding strand of the *PAH* gene in the locust (i.e., sense lncRNA). *PAHAL* positively regulated *PAH* expression and dopamine production in the brain. In addition, *PAHAL* modulated behavioral aggregation of the locust. Mechanistically, *PAHAL* mediated the transcriptional activation

**Data Availability Statement:** All relevant data are within the manuscript and its Supporting Information files.

**Funding:** This work was supported by the National Science Foundation of China (grant number 31920103004 and 31872303 to L.K. and 31872304 to B.C.; www.nsfc.gov.cn), the grants from Chinese Academy of Sciences (QYZDY-SSW-SMC009; www.cas.cn), and Plan for 100 Excellent Innovative Talents of Hebei Province (SLRC2019019; jyt.hebei.gov.cn). The funders had no role in study design, data collection and analysis, decision to publish, or preparation of the manuscript.

**Competing interests:** The authors have declared that no competing interests exist.

of *PAH* by recruiting SRSF2, a transcription/splicing factor, to the promoter-associated nascent RNA of *PAH*. These data support a model of feedback modulation of dopamine biosynthesis and behavioral plasticity via a sense lncRNA in the catecholamine metabolic pathway.

## Introduction

Long noncoding RNAs (lncRNAs) are transcripts comprising > 200 nucleotides and possessing minimal or non-existent protein-coding capacity [1] and are increasingly recognised as key players in numerous cellular processes [2,3]. LncRNAs are the main products of RNA polymerase II, often polyadenylated and processed through splicing [3]. Growing evidence shows that most lncRNAs may be functionally relevant. LncRNAs engage in various biological processes, such as X-chromosome inactivation [4–6], DNA damage response [7,8], differentiation and development [5,9,10], metabolism [11,12] and immunity and disease response [12–14].

The brain and neuronal specificity of lncRNA expression has prompted the exploration of the potential roles of lncRNAs in neuronal development and cognitive and behavioural regulation [10,15–21]. In *Drosophila* species, cytoplasmic lncRNA *yar* regulates sleep behaviour [22]. The lncRNA *CRG* exhibits spatiotemporal specific expression patterns within the central nervous system; in addition, *CRG* affects the locomotor behaviour of *Drosophila* by positively regulating a neighbouring gene that encodes a $Ca^{2+}$/calmodulin-dependent protein kinase [23]. LncRNAs can regulate gene expression through distinct modes [19,24–26], such as their association with transcription factors and/or chromatin modification factors [3,5,27–29]. In mice, *Gomafu* mediates anxiety-like behaviour by maintaining the polycomb repressive complex 1 at the promoter of the schizophrenia-related gene *beta crystallin* [30]. Many more findings implicate the widespread involvement of lncRNAs in neuronal response and neurological diseases [10,17,18,29,31]. Despite the recognised roles of lncRNAs in the neuronal system and behaviour, how neuronal and behavioural responses to environmental stimuli are modulated at the cellular and organismal levels by lncRNAs remains incompletely understood.

The metabolic and signal transduction pathways of dopamine (DA) are important for the behavioural responses of animals. DA is responsible for motor control, learning and memory and is associated with several important neural diseases [32,33]. Phenylalanine hydroxylase (PAH, also known as Henna in *Drosophila* and *Locusta*) catalyses the synthesis of tyrosine from phenylalanine and affects the production of DA and other bioamines in brain [34–37] (Fig 1A). We previously elucidated the metabolic and signalling pathways of DA-regulated aggregation behaviour in the migratory locust *Locusta migratoria*, which exhibits density-dependent behavioural plasticity [38–40]. Gregarious (G) locusts display high mobility and conspecific attraction. By contrast, solitarious (S) locusts are tardy and repulsive to other conspecific individuals [41]. Locusts can reversibly and rapidly shift their behaviours between the S and G phases in response to population density changes. Specifically, G nymphs exhibit significant behavioural solitarization upon 1 h isolation and full solitarization upon 16 h isolation. S nymphs display significant behavioural gregarization upon 32 h crowding [42]. Therefore, locusts are considered ideal models for studying animal behavioural plasticity [38,41,43]. *PAH* expression affects the catecholamine metabolic pathway (e.g., DA synthesis) and controls the behavioural phase shift of locusts [39,40]. Specifically, miR-133 targeting *PAH* coding region repressed *PAH* expression, reduced DA production and drove locust soliterization while activated, and vice versa while inhibited [36]. However, the molecular mechanism underlying the

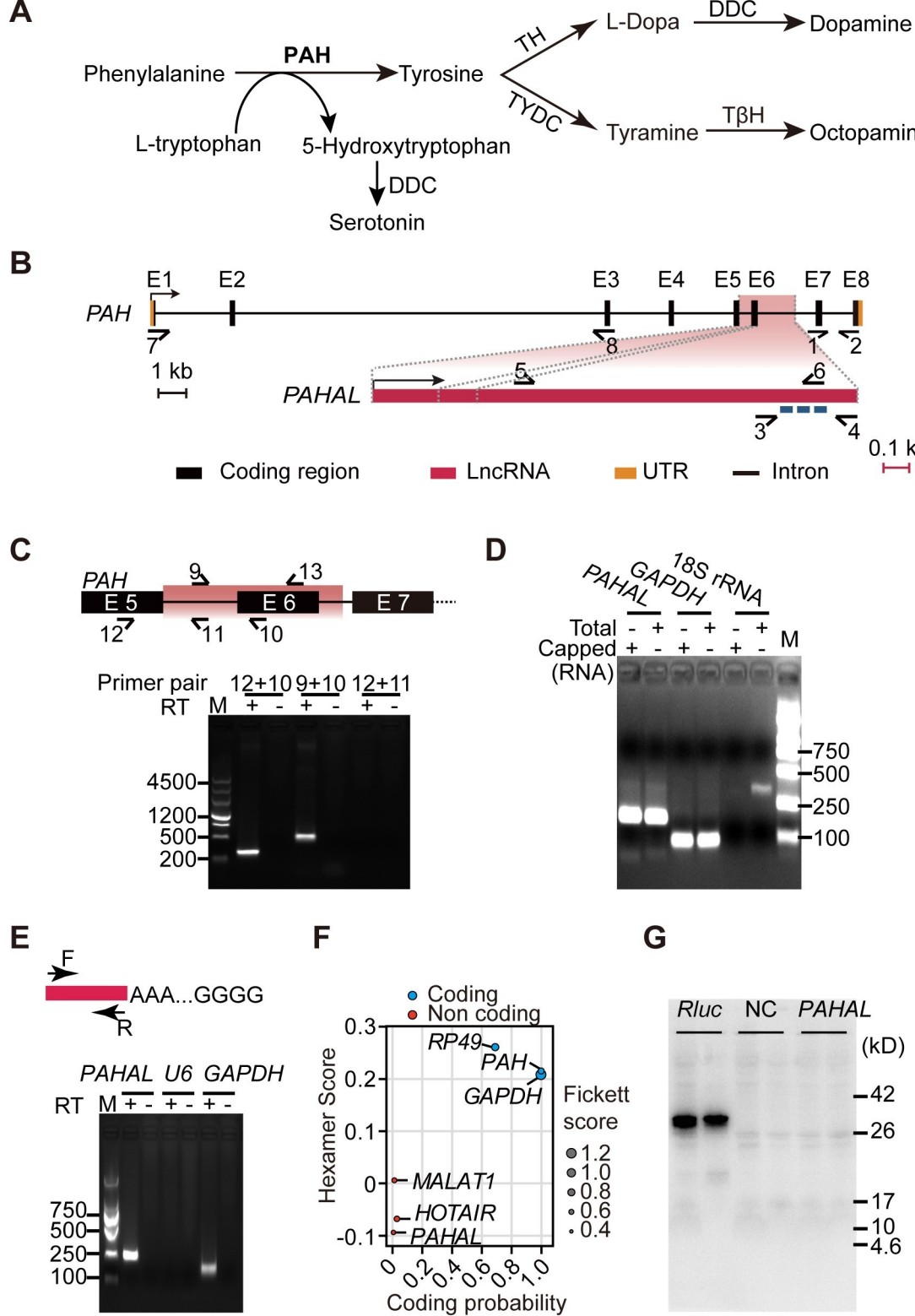

**Fig 1. *PAHAL* is expressed as a sense lncRNA transcript within the *PAH* locus.** (A) Schematic representation of the catecholamine metabolic pathway. PAH catalyses the reaction from phenylalanine to tyrosine. PAH, phenylalanine hydroxylase; TH, tyrosine hydroxylase; TYDC, tyrosine decarboxylase; TβH, tyramine β-hydroxylase; DDC, decarboxylase. (B) Gene and transcript structure of the locust *PAH* and *PAHAL*. The bent arrows indicate the transcriptional orientation of *PAH* and *PAHAL*,

respectively. "E 1–8" represents the eight exons of *PAH* transcript. The half-arrows indicate the following strand-specific primers for quantitative PCR (qPCR): primers 1 and 2 for *PAH*, primers 3 and 4 for *PAHAL*, primers 5 and 6 for *PAHAL* dsRNA, and primers 7 and 8 for *PAH* dsRNA. The broken line represents siRNA sequences used for *PAHAL* knockdown. The entire *PAH* locus is drawn proportional to its length. The black scale bar for *PAH* locus represent 1 kb. The gene structure of *PAHAL* is scaled up in red. The red scale bar for *PAHAL* represent 0.1 kb. (C) *PAHAL* and *PAH* were different transcripts. The half-arrows indicate the following strand-specific primers for PCR: Primer 13 is the specific reverse primer for cDNA synthesis of *PAH* and *PAHAL*, primers 10 and 12 are for *PAH* (289 bp), primers 9 and 10 are for *PAHAL* (476 bp), and primers 11 and 12 yield no PCR product. The red shading indicates the location of *PAHAL* in *PAH* locus. (D) 5′-triphosphate cap of *PAHAL* validated through the 5′ exonuclease digestion assay. *GAPDH* and *18S* rRNA acted as the positive and negative controls, respectively. "Total" means total RNA in reverse transcription (RT), and "capped" indicates capped RNA that was 5′ exonuclease digested in the cDNA synthesis. (E) Detection of *PAHAL* poly(A) tail by RT-PCR in the brain RNA of the fourth-stadium nymphs. Total RNA was G/I-tailed, reverse-transcribed and amplified with gene-specific primers F and R. *U6* and *GAPDH* were used as negative and positive control, respectively. (F) Coding potential of the transcript in the *PAHAL* and reference transcripts according to the CPAT algorithm. In the references, *MALAT1* and *HOTAIR* are lncRNAs, and the others are protein-coding genes. (G) *In vitro* translation assay for *PAHAL*. *Rluc*, biotinylated luciferase; NC, no DNA-template control.

fine-tuning of *PAH* expression for the mediation of brain DA dynamics in behavioural shifts remains ambiguous.

In this study, we explored a new regulatory mechanism, which involves an lncRNA in the DA metabolic pathway and underlies the elaborate control of locust behavioural plasticity. We discovered a novel *PAH*-activating lncRNA (*PAHAL*), which is sense to the ancestral *PAH* gene. *PAHAL* tightly controls the metabolic regulation of DA biosynthesis in the brain and mediates reversible behavioural changes in response to population density. Mechanistic analysis revealed that *PAHAL* can enhance *PAH* expression by recruiting the transcription activator serine/arginine-rich splicing factor 2 (SRSF2) to the *PAH* promoter-proximal region. These results demonstrate that the lncRNA orchestrates gene expression for DA biosynthesis by regulating the local feedback transcription of *PAH*. Thus, our findings provide new insights into the role of lncRNAs in the fine tuning of neuronal and behavioural responses cued by environmental stimuli.

## Results

### Sense lncRNA *PAHAL* is expressed from the intron/exon of the *PAH* gene locus

We identified *PAHAL*, a novel lncRNA that overlaps with the *PAH* gene, in the genome and transcriptomes of the locust. We validated the transcriptional origin of *PAHAL* relative to that of *PAH* through 5′ and 3′ RACE. The locust *PAH* gene encodes PAH by a 1,652 bp-long transcript comprising eight exons (Fig 1B). *PAHAL* is unspliced and is 2,612- nucleotide (nt) long (S1 Fig). The *PAHAL* sequence covers the sixth exon and part of the two introns flanking the exon in the *PAH* gene (Fig 1B and S1 Fig). Sense-specific reverse transcription polymerase chain reactions (RT-PCRs) confirmed that *PAH* and *PAHAL* are different transcripts (Fig 1C). The 5′ exonuclease digestion assay indicated that *PAHAL* is 5′ capped (Fig 1D), and the RT-PCR from the 3′-end tailed-RNA demonstrated that *PAHAL* is a polyadenylated transcript (Fig 1E). These results indicate that the transcription direction of *PAHAL* is the same as that of the canonical *PAH*, and *PAHAL* transcription is sense to *PAH* gene (Fig 1B).

Open reading frame (ORF) analysis showed that 19 short ORFs are present in *PAHAL*. The longest ORF (201 nt) is located at most 3′ end and annotated as the long terminal repeat region of a *Gypsy* transposable element. No match to any known genomic sequence of insect species, except the contained *PAH* exon six (262 nt), was identified. Therefore, the lncRNA sequence is not conserved among insect species. The coding capacity assessment by using the Coding Potential Assessment Tool (CPAT) [44] showed that *PAHAL* lacks protein-coding capacity (Fig 1F). An *in vitro* translation assay demonstrated that no protein was produced by *PAHAL* expression (Fig 1G). Therefore, *PAHAL* is a sense noncoding RNA generated from the *PAH* locus.

## *PAHAL* controls dopamine biosynthesis by regulating the *PAH* expression

We knocked down *PAHAL* through dsRNA (double-stranded RNA) and siRNA (small interfering RNA) interferences to test the regulatory relationship between *PAHAL* and *PAH* in the G locust brain. The dsRNA and siRNAs of *PAHAL* were targeted to the right intron flanking the sixth exon in the *PAH* gene and ought to be specific for *PAHAL* not influence the pre-mRNA of *PAH*. Primer sequences of these dsRNAs and siRNAs can be found in S3 Table. *PAHAL* dsRNA injection in the brain significantly repressed the *PAHAL* expression (*t* test: $P = 0.002$, N = 9). The *PAHAL* knockdown reduced the *PAH* mRNA level by 91% (*t* test: $P = 0.006$, N = 9) and *PAH* protein level by 96% (*t* test: $P < 0.001$, N = 4; Fig 2A). The *PAHAL*-specific knockdown through siRNA interference provided similar results (Fig 2B). However, *PAH* knockdown had no effect on *PAHAL* level (Fig 2C). These findings imply that *PAHAL* is involved in the positive regulation of *PAH* in locust brains.

Thereafter, we applied brain transcriptome sequencing to investigate the downstream pathways affected by the *PAHAL* expression. The RNA-seq data showed that 52 genes were significantly altered by *PAHAL* knockdown (Fisher's exact test: $P < 0.05$, fold change $> 2$, N = 3; Fig 2D and S1 Table, Supplemental File 1). *PAH* was the most differentially expressed gene (Fisher's exact test: $P = 2.5 \times 10^{-37}$, N = 3). The functional clustering of the 52 genes revealed that the phenylalanine metabolism pathway to include the *PAH* gene was the most significantly enriched pathway (Fisher's exact test: $P = 4.9 \times 10^{-4}$, N = 3; Fig 2E).

*PAH* encodes a dual function enzyme, that hydroxylates both phenylalanine for tyrosine synthesis and tryptophan for the synthesis of 5-hydrotryptophan (5-HTP), the precursor of serotonin in insects [34,45,46]. Therefore, we then quantified the levels of tyrosine, a direct product of PAH catalysis, and the six downstream metabolites in the catecholamine metabolic pathways, namely, L-dopa, DA, 5-hydroxytryptophan, serotonin, tyramine and octopamine by liquid chromatography–mass spectrometry (LC–MS) (Figs 1A and 2F). The *PAHAL* knockdown significantly decreased the concentrations of tyrosine (*t* test: $P < 0.001$, N = 10), and metabolites (i.e. L-dopa, *t* test: $P = 0.001$, N = 8) and DA (*t* test: $P < 0.001$, N = 7), in one downstream branch pathway but not those (i.e. tyramine and octopamine) in another one in the brain. Such knockdown also reduced the content of 5-hydroxytryptophan (*t* test: $P < 0.001$, N = 10) but not that of the downstream product, such as serotonin (Fig 2F). These results indicate that *PAHAL* affects the DA synthesis in the catecholamine metabolic pathway.

## *PAHAL* is a positive regulator of *PAH* expression in locust behaviour

We tested the functional relationship between *PAHAL* and *PAH* by monitoring their expression levels in five tissues from G nymphs (Fig 3A). *PAHAL* and *PAH* were highly expressed in the locust brain. G locusts display high mobility and sociable. By contrast, S locusts are sedentary and live at very low densities [41]. The G and S individuals possess distinct, population density-dependent behavioural features [38]. Thus, we compared the expression levels of the two transcripts in the brains of G and S locusts (Fig 3B). The expression of *PAHAL* and *PAH* in the G locusts were 17.7- (*t* test: $P = 0.004$, N = 5) and 2.2-fold higher (*t* test: $P < 0.001$, N = 7), respectively, than those in the S locusts (Fig 3B). Accordingly, *PAHAL* and *PAH* were upregulated in the G locust brain. The brains of G locusts were subjected to fluorescence in situ hybridisation (FISH). The result showed that *PAH* and *PAHAL* transcripts are both localised to neuronal cell bodies, but as expected *PAH* is cytoplasmic whereas *PAHAL* is nuclear (Fig 3C and S2 Fig). Furthermore, *PAH* and *PAHAL* have broad expression patterns rather than being localised only in a subset of locust neurons.

We tested the time-course expression dynamics of *PAHAL* and *PAH* in the brain to determine their dynamic correlations during locust isolation and crowding treatments (Fig 3D).

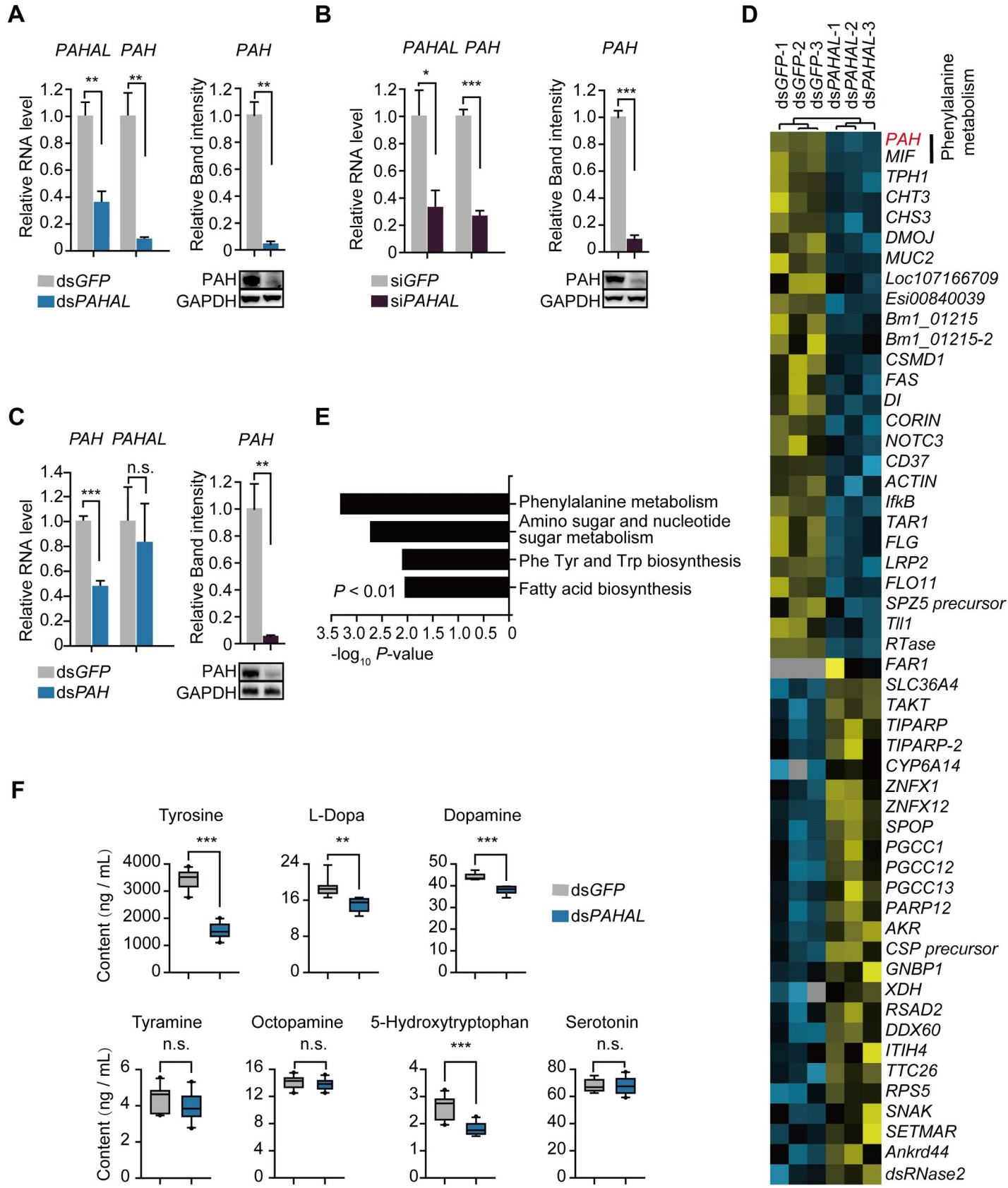

**Fig 2. *PAHAL* regulates the *PAH* expression and controls the DA biosynthesis in the locust brain.** The *PAHAL* and *PAH* expression levels in the locust brain at 72 h after dsRNA (A) and siRNA (B) knockdown of *PAHAL*. ds*GFP* and si*GFP* represent *GFP* dsRNA and siRNA injection and acts as the control. ds*PAHAL* and si*PAHAL* represent *PAHAL* dsRNA and siRNA injection. The expression levels in ds*PAHAL* and si*PAHAL* are normalised against those in ds*GFP* and si*GFP*. *Rp49* is the internal control. Nine biological replicates of 10 brains were measured. (C) *PAH* knockdown through ds*PAH* injection did not alter the level of *PAHAL*. Five biological replicates of eight brains were measured. (D) Median-centered, hierarchical clustering of annotated genes that were altered in *PAHAL*-silenced brains. Three biological replicates of 12 brains were sequenced. (E) Functional clustering of differentially expressed genes using KEGG enrichment analysis. Only $P < 0.05$ terms are shown (Fisher's exact test). (F) Effects of *PAHAL* knockdown on the bioamine metabolite level in the catecholamine metabolic pathway in the brain. Bioamine metabolite levels were measured through high performance liquid chromatography–mass spectrometry. Ten biological replicates of 12 brains were measured. The data in this figure, as well as those in the following figures, are represented as the mean ± SEM with the fourth-stadium nymphs, unless stated otherwise. Student's *t*-test: $^{*}P < 0.05$; $^{**}P < 0.01$; $^{***}P < 0.001$.

*PAHAL* was significantly upregulated at 4 h upon aggregation (*t* test: $P = 0.002$, N = 8) compared with that at 0 h. The upregulation of the *PAHAL* expression was sustained at 8 h (*t* test: $P = 0.040$, N = 8). By contrast, the *PAHAL* expression was significantly downregulated at 4 h after isolation (*t* test: $P = 0.013$, N = 8). Such expression continued to decrease at 8 h after isolation (*t* test: $P = 0.001$, N = 8). *PAH* exhibited the same time-course expression pattern. Thus, the expression levels of *PAHAL* and *PAH* are positively correlated during locust aggregation and isolation.

## *PAHAL* controls phase-related behavioural transition

We investigated the role of *PAHAL* in behavioural regulation in the locusts using dsRNA and siRNA interference of *PAHAL*, respectively. Locust aggregation in an arena was measured by using an established automatic behavioural assay [40,42] (Fig 4A). Aggregation behaviour was quantified by using $P_{greg}$, a summary statistic provided by this behavioural assay and a reliable indicator of locust aggregation propensity [40]. The shift in the median $P_{greg}$ value of G locusts from 0.89 to 0.23 when *PAHAL* was knocked down through dsRNA interference indicated a significant behavioural shift from the G to the S states (Mann–Whitney *U* test: $P < 0.001$, $N_{dsGFP} = 30$, $N_{dsPAHAL} = 57$, Fig 4B). The results from siRNA interference lead to a change of $P_{greg}$ from 0.90 to 0.20 (Mann–Whitney *U* test: $P < 0.001$, $N_{siGFP} = 33$, $N_{siPAHAL} = 35$) (Fig 4C). This outcome from siRNA demonstrates an effect of *PAHAL* on locust aggregation similar to that from dsRNA.

Furthermore, we examined the specific parameters of the behavioural traits of locusts after the gene expression interference. In comparison with the ds*GFP* control treatment, the *PAHAL* knockdown reduced the total distance of movement by 71% (*t* test: $P < 0.001$, $N_{dsGFP} = 30$, $N_{dsPAHAL} = 57$), decreased the total duration of movement by 95% (*t* test: $P < 0.001$, $N_{dsGFP} = 30$, $N_{dsPAHAL} = 57$) and induced the reversion from attraction to repulsion, as shown by the attraction index (Mann–Whitney *U* test: $P = 0.008$, $N_{dsGFP} = 30$, $N_{dsPAHAL} = 57$; Fig 4D). We also examined the specific behavioural effect from siRNA knock down. The *PAHAL* interference by siRNA has behavioural effects similar to that by dsRNA in terms of mobility and conspecific attraction (Fig 4E). Overall, *PAHAL* effectively modulates the behavioural aggregation of the locusts.

## *PAHAL* facilitates the transcription activation of *PAH* by interacting with the *PAH* promoter-proximal region

We determined the mechanism by which *PAHAL* mediates *PAH* transcription. The sequence analysis revealed that one nuclear localisation signal (NLS) [47] is present at the 3′ end of *PAHAL* but not in *PAH* (S3 Fig). The nuclear fractionation experiment showed that 94% of *PAHAL* mRNAs (possibly together with trace amount of *PAH* pre-mRNA) localises in the nucleus relative to nuclear RNA *U6* (positive control) and cytoskeleton actin (negative control). By contrast, 97% of *PAH* mRNAs localised in the cytoplasm (Fig 5A). FISH results also

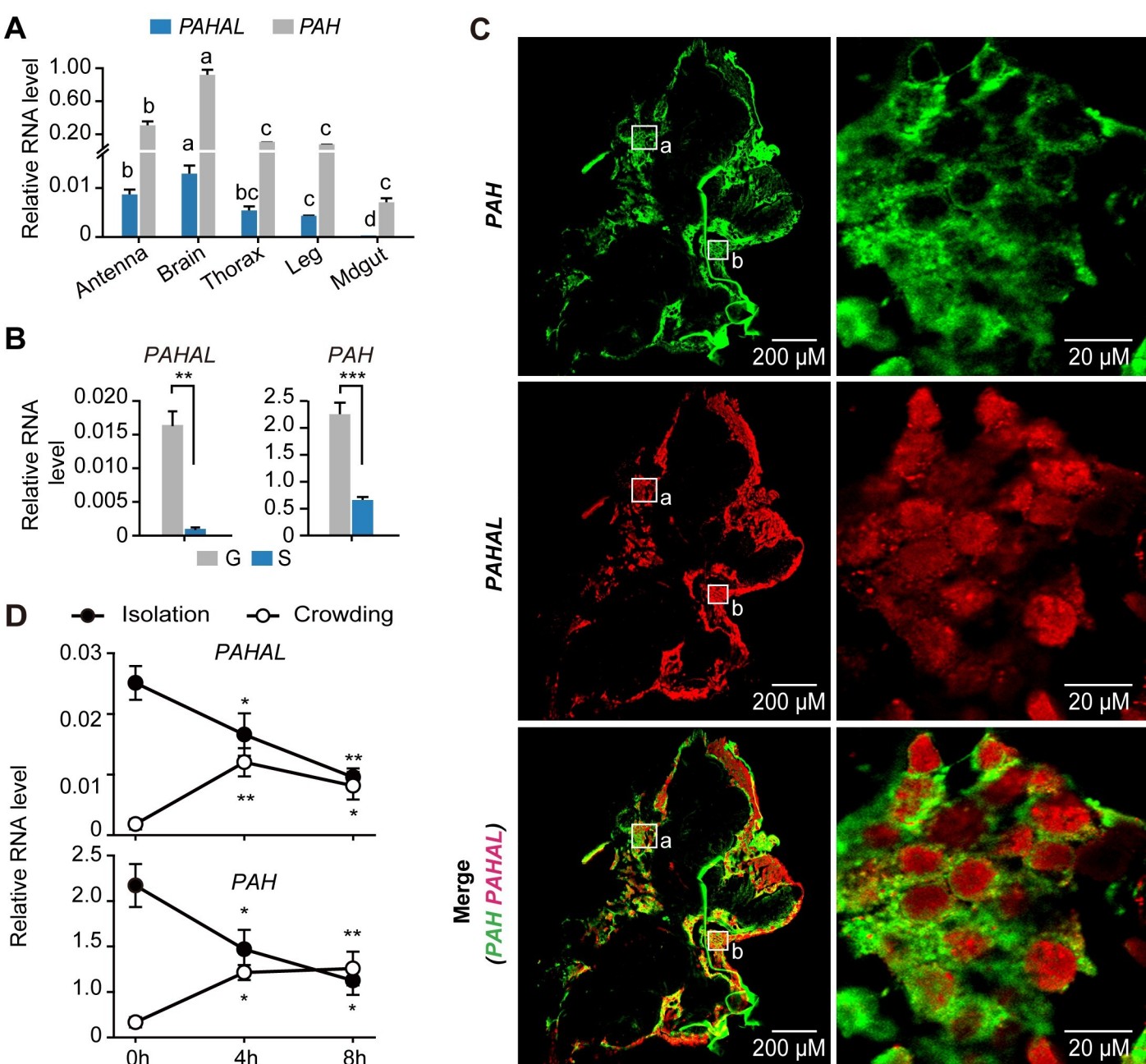

**Fig 3. *PAHAL* expression is positively correlated with the *PAH* expression in the brain during the behavioural transition of locusts.** (A) Tissue expression of *PAHAL* and *PAH*. The mRNA level was quantified through qPCR. Means labelled with different letters within each gene are significantly different (*P* < 0.05). (B) *PAHAL* and *PAH* expression in the brains of gregarious (G) and solitarious (S) locusts. (C) Localisation of *PAHAL* and *PAH* in the G brains as revealed through double fluorescence in situ hybridization (FISH). Images are shown at 10× (left images) and 63× (right images) magnification. The "a" squares specifically indicate the brain region of the right images in the locust protocerebrum. The brain region of the Fig 5B is indicated by the "b" squares in the locust deutocerebrum. (D) Profiles of the *PAHAL* and *PAH* expression in the brain during locust isolation and crowding. Eight biological replicates of eight brains were measured. Asterisks indicate significant differences between each time point and at 0 h (*P* < 0.05). Student's *t*-test: *P* < 0.05; **P* < 0.01; ***P* < 0.001.

showed that *PAHAL* and *PAH* localised in the nucleus and cytoplasm, respectively (Figs 3C and 5B). This finding indicates that these two transcripts are distinct in the subcellular location.

Whether *PAHAL* regulates *PAH* expression through a *cis* interaction with a *PAH* promoter was determined. The sequence from −1,169 nt to +89 nt relative to the *PAH* initiator was

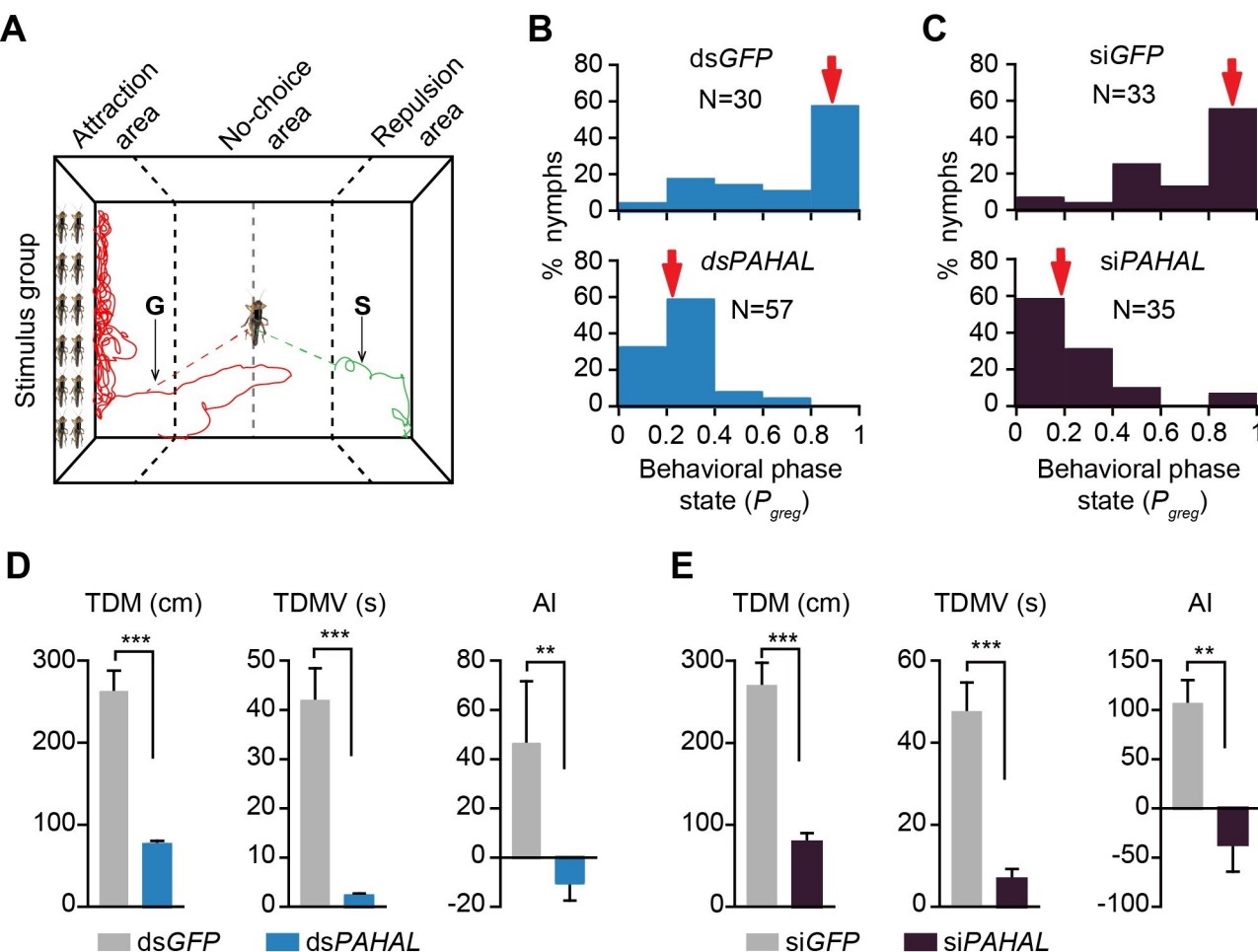

**Fig 4. *PAHAL* modulates locust behavioural aggregation.** (A) Movement trajectories of the G and S nymphs in the behavioural assay arena. A stimulus group of 30 G nymphs were placed behind a transparent porous partition on the left. (B) dsRNA interference of the *PAHAL* expression resulting in the behavioural transition from G to S phase. $P_{greg}$ is a measure of the predisposition of locusts to swarming, that is, 0 for typical solitarious behaviour and 1 for typical gregarious behaviour. Median $P_{greg}$ values are indicated by red arrows. (C) Behavioural changes resulting from siRNA interference of *PAHAL*. (D) dsRNA knockdown of the *PAHAL* expression altering behavioural traits, including the total distance moved (TDM), the total duration of movement (TDMV), and the attraction index (AI). (E) Behavioural traits affected by the siRNA knockdown of *PAHAL*. Asterisks indicate the significance level by Student's *t*-test for TDM and TDMV and by Mann–Whitney *U* test for AI: $^{**}P < 0.01$; $^{***}P < 0.001$.

selected as the regulatory segment of the *PAH* promoter (labelled as P+5′-UTR). The luciferase assay showed that *PAHAL* substantially enhanced the luciferase expression driven by the *PAH* promoter in S2 cells (*t* test: $P < 0.001$, N = 4). By contrast, *PAHAL* expressed in the reverse orientation (i.e., reverse *PAHAL*) had no effect on *PAH* promoter activity (Fig 5C). Thus, *PAHAL* enhances the promoter activity of *PAH* not by acting as an enhancer element.

Moreover, the specific regulatory region in the *PAH* promoter targeted by *PAHAL* was determined by testing the transcription activities of truncated promoters coexpressing *PAHAL*. *PAHAL* significantly enhanced the activity of these promoter regions, even that of the shortest promoter (from −48 nt to +89 nt; *t* test: $P < 0.001$, N = 4; Fig 5D). Thus, the active *PAHAL* interacting region is close to the transcription start site (TSS) of *PAH*. The removal of the 5′-UTR abrogated the *PAHAL*-mediated transcriptional activation (Fig 5E). Therefore, *PAHAL* promotes the *PAH* expression by actively interacting with the promoter-proximal region, including the 5′-UTR, of the *PAH* gene.

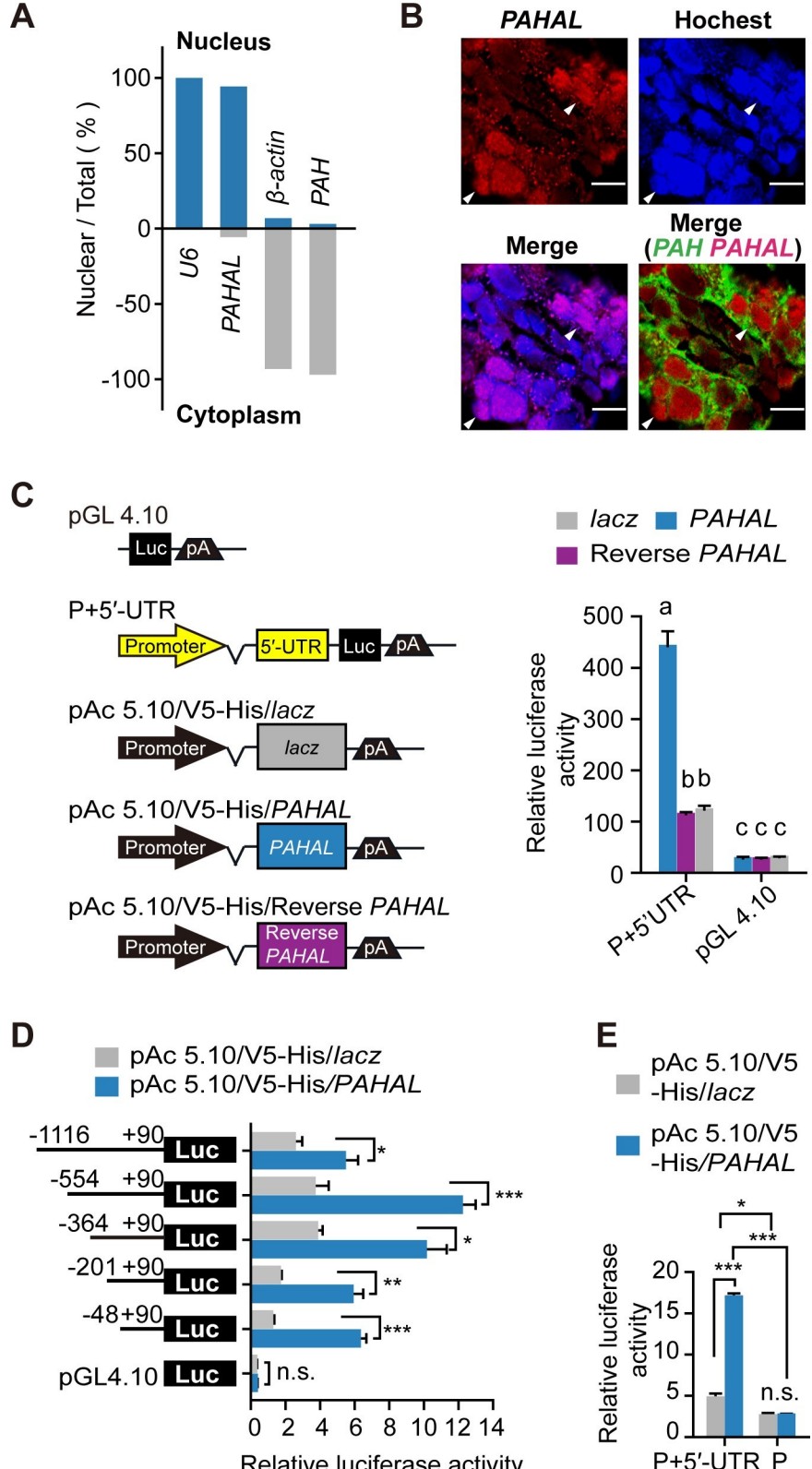

**Fig 5. Nuclear-enriched *PAHAL* promotes the *PAH* expression by interacting with the promoter-proximal region of *PAH*.** (A) Subcellular localisation of *PAHAL* and *PAH* in the brain. Five biological replicates of eight brains were

examined for each treatment. (B) FISH detection of the *PAHAL* probe (red), *PAH* probe (green), and Hoechst 33342 (blue) in the brain. The arrows indicate the subcellular location of *PAHAL* (in the nucleus) and *PAH* (in the cytoplasm). Scale bars, 50 μM. The brain region in these images was indicated by the "b" squares of Fig 3C. (C) *PAHAL*-enhanced *PAH* promoter activity. The *PAH* promoter construct (P+5′-UTR) contained −1,168 to +89 bp relative to the TSS. "P" represents −1,168 to +1 bp, and "5′-UTR" denotes +1 to +89 bp. Reverse *PAHAL* is a reverse full-length fragment of *PAHAL*. *Lacz* is a fragment of *Lacz* gene containing frame shift and acted as a negative control. *PAHAL*, reverse *PAHAL* and *lacz* were constructed into pAC5.1/V5-His A vectors to their high-level transient expression in *Drosophila* S2 cells. (D) *PAHAL* interacts with the proximal promoter of *PAH*. (E) 5′-UTR of *PAH* is required for transcriptional activation by *PAHAL*. Error bars indicate ± SEM. Student's *t*-test: $^*P < 0.05$, $^{**}P < 0.01$; $^{***}P < 0.001$; n.s., insignificant.

## *PAHAL* regulates *PAH* expression by binding with SRSF2

To determine proteins that interact with *PAHAL*, we performed RNA pulldown with brain tissue extracts (Fig 6A) and identified 11 protein candidates by mass spectrometry (S2 Table). Among these proteins, SRSF2 bound with the sense, but not antisense, strand of *PAHAL* (Fig 6B) and was associated with behavioural phase changes (Fig 6C and S4 Fig). Specifically, *SRSF2* expression in the brain of a G locust significantly differed from that of an S locusts (*t* test: $P < 0.001$, N = 8). *SRSF2* was continuously upregulated during locust crowding treatments and downregulated during isolation treatments (Fig 6C). Thus, the *SRSF2* expression is positively related to the *PAHAL* and *PAH* expression during locust behavioural transition.

To determine the physical interaction between *PAHAL* and SRSF2 that binds to the promoter-associated nascent RNA during transcription regulation, we conducted several *in vitro* and *in vivo* RNA immunoprecipitation (RIP). *In vitro* RIP using an antibody against V5 tag in SRSF2-V5 protein-overexpressed cell lysis revealed significant *PAHAL* enrichments with SRSF2 (S5A Fig and Fig 6D, *t* test: $P = 0.010$, N = 6). *In vivo* RIP with brain tissues showed that endogenous *PAHAL* mRNA and SRSF2 were significantly enriched by the SRSF2 antibody (*t* test: $P = 0.004$, N = 5 for enriched *PAHAL*, Fig 6D; *t* test: $P = 0.029$, N = 5 for SRSF2, S5B Fig). Thus, the SRSF2 protein physically binds with *PAHAL* RNA.

Thereafter, we examined the regulatory effects of SRSF2 on *PAHAL*-mediated transcriptional activation through the SRSF2 knockout and overexpression. We first used SRSF2 protein-depleted cells generated from mouse embryonic fibroblasts (SRSF2-MEFs); an HA-tagged SRSF2 gene replaced the endogenous gene and expressed from a tetracycline (tet)-off promoter, thereby enabling the elimination of the protein by adding the tet analogue DOX [48,49]. The *PAH* promoter activity was significantly inhibited by 67% by SRSF2 knockout (One-Way ANOVA: $P < 0.001$, N = 6). The enhanced activity from co-transfected *PAHAL* was abrogated (One-Way ANOVA: $P < 0.001$, N = 6) by SRSF2 knockout (Fig 6E). By contrast, the SRSF2 overexpression in S2 cells significantly increased the *PAH* promoter activity by 56% (One-Way ANOVA: $P < 0.001$, N = 5). The elevated promoter activity was further enhanced by 91% (One-Way ANOVA: $P < 0.001$, N = 5) with *PAHAL* co-transfection. The effects of the SRSF2 and *PAHAL* expression exhibited a significant interaction in the *PAH* promoter activation (Mann–Whitney *U* test: $P < 0.001$, N = 5, Fig 6F). Therefore, SRSF2 substantially promoted the *PAHAL*-mediated *cis*-activation effects.

## *PAHAL*–SRSF2 complex binds with sequence specificity for transcriptional regulation

We determined the specific sites for SRSF2 interaction in the *PAHAL* sequence. The domain mapping of sequence fragments containing the 3′ end of *PAHAL* revealed that all the fragments interacted with SRSF2 (Fig 7A, left panel). The mapping analysis of sequence fragments covering the 5′ end of *PAHAL* revealed that only the fragment that simultaneously covered the

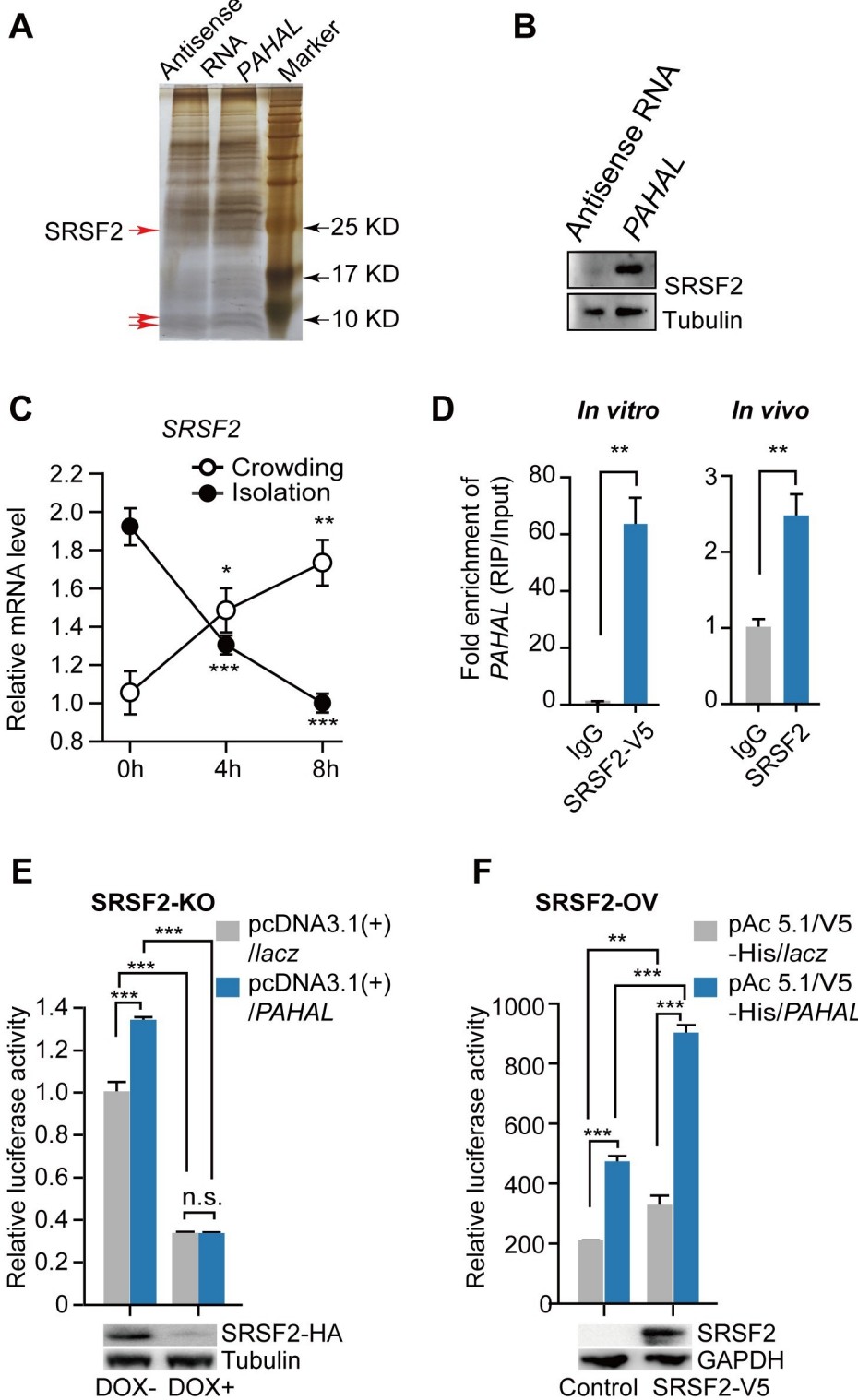

**Fig 6.** *PAHAL* **activates** *PAH* **transcription by interacting with SRSF2.** (A) RNA pulldown identifying *PAHAL*-associated proteins in the brain. Red arrows represent different protein bands between *PAHAL* and its antisense. (B) Western blot detects SRSF2 in protein pools from *PAHAL* RNA pulldown (N = 4). (C) *SRSF2* expression in the brain during locust isolation and crowding. Eight replicates of eight brains were measured. (D) RNA immunoprecipitation (RIP) enrichment verifies that SRSF2 bound with *PAHAL in vitro* and *in vivo*. Five replicates of eight brains were measured. (E, F) SRSF2 facilitates the *PAHAL*-mediated transcription activation of *PAH*. Luciferase assays of P+5′-

UTR were performed by using SRSF2 protein-depleted mouse embryonic fibroblasts (SRSF2-MEFs, SRSF2-KO). "DOX +" represents SRSF2 knockout, whereas "DOX−" indicates the normal expression of SRSF2 in SRSF2-MEFs or in either S2 cells overexpressed SRSF2 (S2, SRSF2-ov). Student's *t*-test: *$P < 0.05$; **$P < 0.01$; ***$P < 0.001$; n.s., insignificant.

3′ end (i.e., the last 245 nt) of *PAHAL* bound with SRSF2 (Fig 7A, middle panel). Two overlapping fragments at the 3′ terminal of *PAHAL* spanning 1,968–2,613 exhibited SRSF2 binding (Fig 7A, right panel). Thus, the sequence at the most 3′ end of *PAHAL* is necessary for *PAHAL*–SRSF2 binding.

Furthermore, we analysed the mechanism by which *PAHAL* and SRSF2 interaction facilitated the transcription activation of the *PAH* gene. SRSF2 facilitates transcription by binding to a high-affinity binding site (exonic-splicing enhancer, i.e., ESE) of nascent RNA [49]. We found three ESE motifs in the 5′-UTR of *PAH* (S5C Fig). The *PAHAL*-mediated activation of the *PAH* promoter activity was disrupted by the mutation of the three ESEs; by contrast, a single ESE mutation did not eliminate the activation effect of *PAHAL* (Fig 7B). Therefore, the

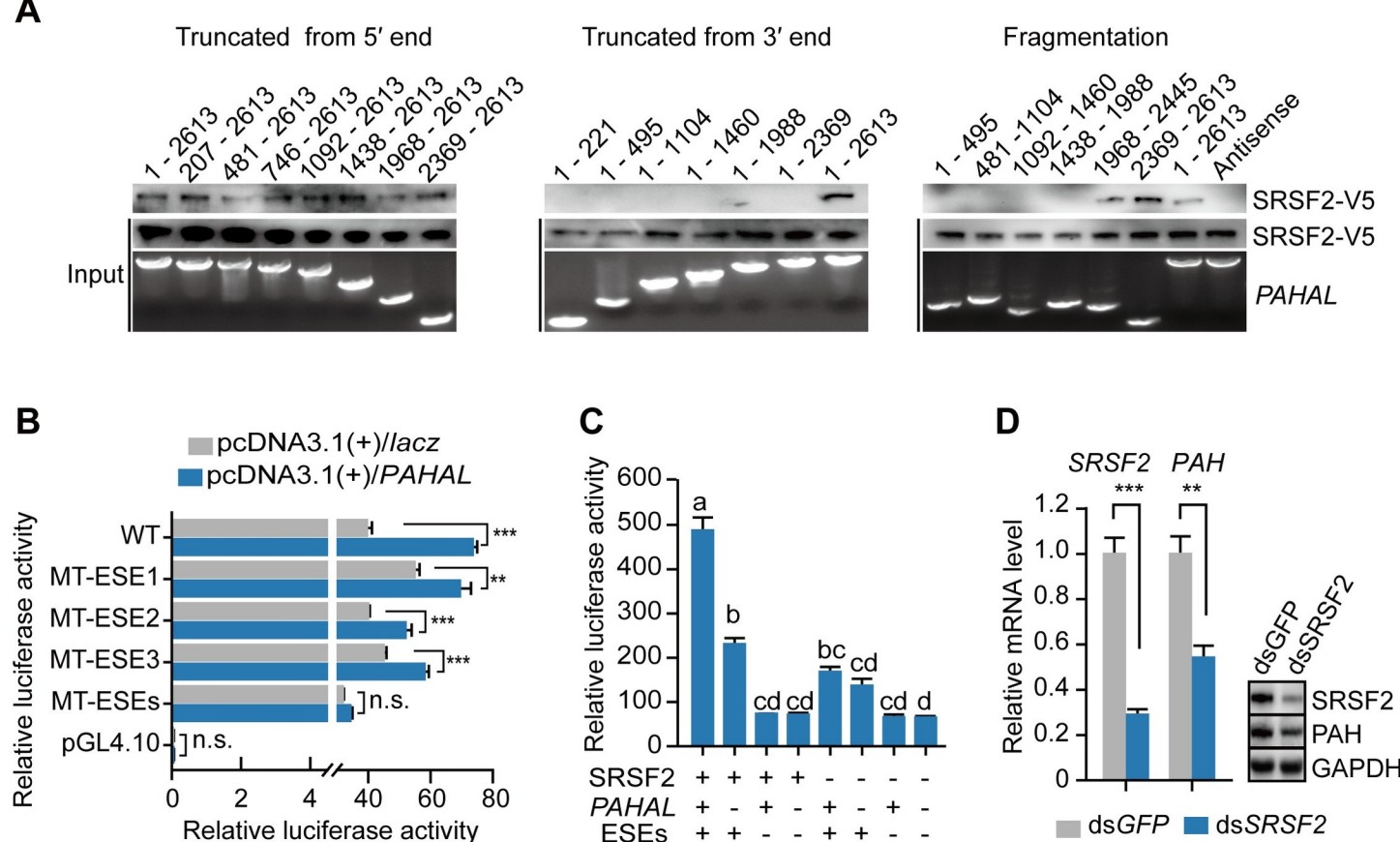

**Fig 7. SRSF2 is recruited by the 3′ terminal of *PAHAL* to *PAH* 5′-UTR for the transcriptional activation of *PAH*.** (A) Mapping analysis of the biotin-labelled fragments of *PAHAL* incubated with SRSF2-MEF lysates that overexpress the V5-tagged SRSF2 (SRSF2-V5) of locusts. (B) Mutational analysis of the activity of ESEs in *PAH* gene with the overexpressed *PAHAL* by using the luciferase assay. WT indicates wild type. MT-ESE1, MT-ESE2 or MT-ESE3 indicate mutated ESE1, ESE2 or ESE3 in the pGL4.10 vector, respectively. MT-ESEs indicate the mutation of all the three ESEs. (C) The interaction of *PAHAL*, SRSF2 and ESE promotes the transcription activation of *PAH*. Means labelled with different letters are significantly different ($P < 0.05$). The following vectors were transfected in SRSF2-MEFs: "ESE+", P+5′-UTR (containing −554/+89 fragment without ESE mutation) in pGL4.10; "ESE−", P+5′-UTR containing the mutated tandem ESEs; "*PAHAL*+", the pcDNA3.1(+)/*PAHAL*; "*PAHAL*−", the pcDNA 3.1(+)/*lacz*; "SRSF2+", the pcDNA 3.1/V5-His/*SRSF2*; "SRSF2−", the pcDNA3.1/V5-His /*lacz*. (D) *SRSF2* knockdown by dsRNA injection reduces the *PAH* expression in the brain. Six replicates of eight brains were measured. Student's *t*-test: *$P < 0.05$; **$P < 0.01$; ***$P < 0.001$; n.s., insignificant.

regulatory function of *PAHAL* involves interaction with the ESEs of the *PAH* 5′-UTR. The luciferase assay further demonstrated that the three factors, namely, *PAHAL*, SRSF2 and tandem ESEs, had significant interactions in the effects of transcriptional regulation (One-Way ANOVA: $P < 0.001$, N = 6). The tandem ESEs exhibited the greatest effects among the three factors (Fig 7C). These results indicate that *PAHAL* directs SRSF2 to the ESEs of *PAH* 5′-UTR to promote *PAH* transcription.

Finally, we validated the role of SRSF2 in the transcriptional activation of *PAH* through RNAi. *SRSF2* knockdown significantly reduced the mRNA level of *PAH* (*t* test: $P = 0.006$, N = 6) and protein level of PAH by 89% (*t* test: $P = 0.037$, N = 3, Fig 7D). Thus, SRSF2 regulates *PAH* transcription *in vivo*.

## Discussion

In our study, we have identified a novel sense lncRNA named *PAHAL* in the *PAH* locus in the catecholamine metabolic pathways, and have revealed its specific role in regulating DA biosynthesis and locust behavioral aggregation. Moreover, we have also characterized the mode of transcriptional activation orchestrated by this lncRNA in the biosynthetic regulation. *PAHAL* exhibits a distinct mode of *cis*-regulation transcription, which is verified in locust, fruit fly and mice *in vivo* and *in vitro*. DA has been confirmed as a crucial modulator for locust behavioural phase transition [36,40]. Thus, our present study established the novel linkage between genetic and epigenetic modulation of behavioural plasticity of locust regulated by DA.

*PAHAL* knockdown attenuated the *PAH* expression and caused a behavioural shift toward S traits. Furthermore, PAH catalyses the reaction from phenylalanine to tyrosine and affects the downstream metabolic pathways of several bioamine neurotransmitters. Immunocytochemical studies show that aminergic neurons present wide distribution in the locust brain [50–52]. Specifically, most dopaminergic neurons are in the upper division of the central body [51]. Serotoninrgic neurons distribute 6 groups that innervate the central complex (groups l-5) [50]. Our study indicates that *PAH* and *PAHAL* are expressed in almost all neurons in the locust brain (Figs 3C and 5B). However, the transcriptome sequencing and LC–MS measurements illustrate that *PAHAL* predominantly affects the DA synthetic pathway in the locust brain (Fig 2), although the results do not exclude non-*PAH* players in DA metabolism and *PAH*'s role in other processes. Thus, this study establishes the regulatory role of *PAHAL* in the orchestration of the gene expression and DA metabolism to modulate animal behavioural transition in response to environmental stimuli.

This study also reveals the regulatory relevance of lncRNAs in strictly controlled genetic programs. Animal behaviours are generally deliberately tuned in response to a specific environmental cue. This response requires triggering the precise spatial–temporal regulation of the expression of specific genes or genetic pathways and the production of specific neurotransmitters [33,40]. The mode of *cis*-regulation of *PAHAL* lncRNA exhibits common transcription mechanism, which is verified in locust, fruit fly and mice. Numerous lncRNA loci act as local regulators and influence the expression of nearby genes through *cis*-regulation [24,25]. These lncRNAs with *cis*-regulatory roles may arise from diverse ways, such as from promoters or enhancers, antisense transcripts or from within introns or near TSSs of other host genes [25]. For example, about half fraction of antisense transcripts are noncoding RNAs [25]. In another study, genetic manipulation in mouse cell lines revealed that, among 12 genomic loci that produce lncRNAs, five loci influence the expression of a neighboring gene in *cis* [24]. These studies have revealed the prevalence of lncRNA-mediated *cis* regulation [53,54]. *Cis*-interference or silencing by intragenic lncRNAs is also a common phenomenon in numerous organisms. For example, the antisense *Tsix* RNA that originates from the *Xist* loci inhibits the expression

of the maternal *Xist* allele [6]. By contrast, *COLDAIR*, which is a sense intronic lncRNA, antagonises *FLC* in *Arabidopsis* [55]. Nevertheless, *PAHAL* is mainly derived from the intronic region of *PAH* but also shares its partial sequence with the sixth exon of the coding *PAH* transcript. Such transcription mode does not solely occur to *PAHAL*. For example, the sense lncRNA *GClnc1* is transcribed in large part from the intronic region of the *SOD2* gene, but it also overlaps with the last exon of two *SOD2* transcripts [56]. *PAHAL* and *PAH* are positively regulated in the behavioural change of the locusts. *In vivo* and *in vitro* analyses demonstrated that nuclear-enriched *PAHAL* promoted the *PAH* expression. *PAHAL* is engaged in a regulatory feedback loop that is stimulated by population density changes, by interacting with *PAH* (Fig 8). Thus, *PAHAL* acts as a sense transcriptional activator and represents an lncRNA subclass that participates in local *cis*-regulation.

In line with the behavioural phase changes, *PAHAL* is generated at transcription levels highly contingent upon locust population density. Thus, *PAHAL* transcription is precisely controlled commensurate with the crowding state and phase-related behaviour of locusts. The *PAHAL* transcription then triggers the fine tuning of the *PAH* expression and in turn the downstream biosynthesis of specific neurotransmitters such as the DA that controls the aggregation behaviour of locusts (Fig 8). Therefore, *PAHAL* is dedicated to orchestrating the *PAH* expression and downstream DA production during the population density-dependent behavioural aggregation of locusts. The density-contingent transcription of *PAHAL* could be also crucial for adjusting the DA dosage. A large proportion of lncRNAs in animal models present high spatial–temporal specificity of expression in various biological processes, such as development and behavioural responses [20,57–59]. Overall, these findings highlight the important role of lncRNAs as a regulator of tightly controlled genetic networks.

Interestingly, two distinct regulatory mechanisms for *PAH* expression rendered by two noncoding RNAs have been employed in the locust. On one hand, lncRNA *PAHAL* is induced by locust crowding, positively regulates *PAH* expression and DA production in the brain at the transcriptional level. On the other hand, miR-133 is upregulated by locust isolation, evokes degradation of *PAH* mRNA and inhibits DA production, thus representing a mechanism for post-transcriptional regulation of *PAH* [36]. Both *PAHAL* and miR-133 are involved in the modulation of behavioral aggregation but in contrasting directions. Therefore, the two mechanisms at different regulatory levels may be orchestrated in response to population density to ensure the fine tuning of *PAH* expression and DA biosynthesis and thus guarantee the deliberate behavioral response of locusts.

*PAHAL* mediates regulatory activation by recruiting the transcription activator SRSF2 to the promoter-proximal region of *PAH*. In this regulatory network (Fig 8), *PAHAL* involves a distinct regulatory element in the lncRNAs–SRSF2 interaction complex that is destined to the transcriptional machinery of the focal gene. Previous studies showed that some lncRNAs participate in regulatory complexes through different mechanisms, such as interaction with transcription factors and chromatin modifiers [44,60]. SRSF2 was first characterised as a splicing factor [49,61–63]. The nuclear-retained lncRNA *MALAT1* is also involved in the regulation of alternative splicing by interacting with SRSF2 [64]. In contrast with *MALAT1*, *PAHAL* functions in transcriptional activation by binding with SRSF2. Notably, SRSF2 also plays an active role in transcription elongation and activation [49,62]. In this study, *PAHAL* RNA recruits SRSF2 to the promoter-proximal region of the target gene to promote local transcription (Fig 8). Therefore, SRSF2 possibly exerts different functions by interacting with various lncRNAs. SRSF2 bound to the ESE sites of the promoter-associated nascent RNA. Three tandem ESE motifs in the 5′-UTR of *PAH* significantly increases the *PAHAL*-mediated activation of the *PAH*-promoter activity in the presence of SRSF2. The ESE sites near the 5′ end of nascent RNA may act as critical signals for transcription activation by triggering the progressive release

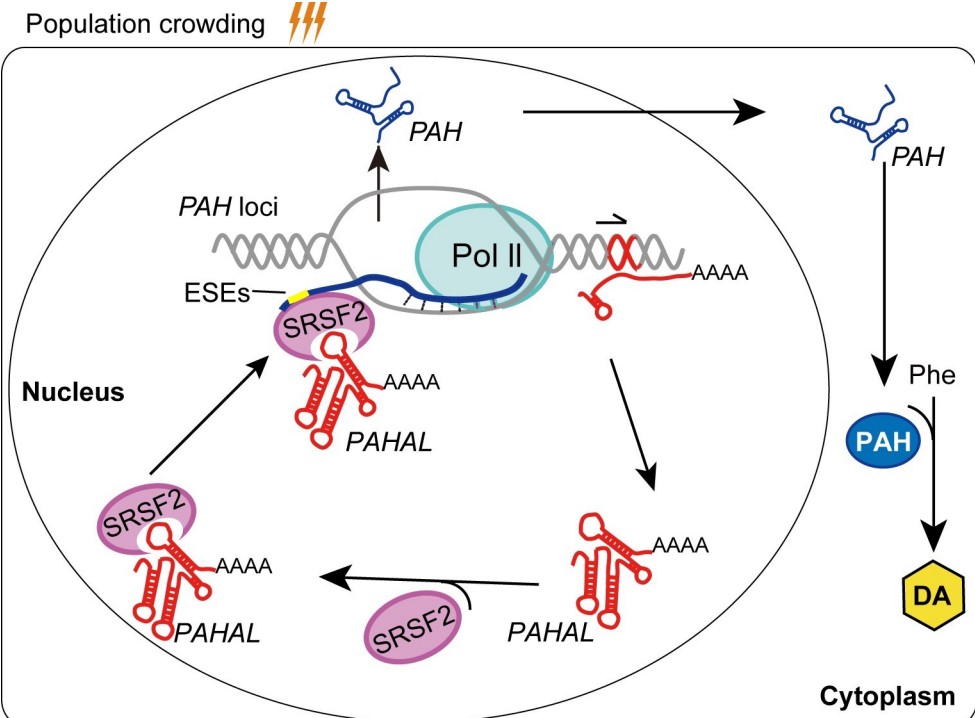

**Fig 8. Working model for the feedback activation of *PAH* via *PAHAL*–SRSF2 interaction in the DA metabolic pathway.** The sense nuclear lncRNA *PAHAL* is transcribed from the exon–intron region of *PAH* gene locus cued by population crowding. *PAHAL* then binds with SRSF2 at its 3′ end to form a RNA–protein complex. *PAHAL* guides the *PAHAL*–SRSF2 complex to bind with the ESE elements of the nascent *PAH* RNA and activates the *PAH* transcription. PAH catalyses phenylalanine into DA. The red genomic region in the *PAH* loci represents the *PAHAL* emanative fragment. The half arrowhead indicates the transcriptional direction of *PAHAL*.

of SRSF2 and P-TEFb from the transcription pause complex [49]. Thus, *PAHAL* may enhance the transcription activation of the *PAH* gene by facilitating SRSF2 binding to the nascent *PAH* RNA. However, the activity assay with reverse *PAHAL* demonstrated that the enhancer effect rendered by *PAHAL* is not due to a genomic enhancer element residing in its sequence (Fig 5C).

The local lncRNA-interwired transcription machinery could orchestrate locus-specific transcription regulation in response to an environmental stimulus. SRSF2 is a general transcription activator in the transcription elongation complex and targets a large set of genes [62]. The locus-specific sense lncRNA *PAHAL* is generated in response to the environmental signal and destined to the transcriptional machinery of the ancestral gene in collaboration with nascent RNA-bound SR proteins (e.g. SRSF2). The lncRNA–SRSF2 interaction complex may have also delivered conformation modifiers to its target genomic loci while still attached to the elongating RNAP II [28,60]. Hence, *PAHAL* may facilitate the regulatory role of SRSF2 in a gene locus-specific manner. The sequence at the 3′ end of *PAHAL* is crucial for the binding of SRSF2 with *PAHAL*, although the underlying mechanism needs further investigation. This region also harbours one NLS that determines the nuclear retention of *PAHAL*. SRSF2 is the only SR protein retained in the nucleus [65]. *MALAT1* to bind with SRSF2 is required for the correct localisation of SRSF2 to nuclear speckles [64,66]. The sequence at the 3′ end of *PAHAL* may confer a structural component or binding site that promotes the nuclear retention of SRSF2. In summary, these data suggest that *PAHAL* could govern the spatial specification of the regulatory components of transcriptional machinery in cells and control the target

specificity of interacting transcriptional factors in animal behavioural response. Such a precise regulatory mechanism derived from interactions between *PAHAL* and SRSF2 enlightens us to develop the novel approaches for behavioural manipulation of animals.

## Materials and methods

### Animals

Locusts were derived from one locust population (Hebei, China) and maintained under standard conditions at the Institute of Zoology, Chinese Academy of Sciences, Beijing, China [67]. The locust colonies were reared under a photoperiod regime of 14 h of light:10 h of darkness at 30˚C ± 2˚C and fed on fresh wheat seedlings and bran. The G locusts were reared in large cages (40 cm × 40 cm × 40 cm) at a density of approximately 500 insects per cage. The S locusts were obtained from the G colony and individually cultured in white metal boxes (10 cm × 10 cm × 25 cm) supplied with fresh air. The S locusts were reared in isolation for at least three generations prior to experimentation. One-day-old fourth-stadium nymphs, which are most prominent for the behavioural phase plasticity [40,67,68], were used in all experiments.

### Cells

*Drosophila* S2 cells (Gibco, NY, USA, R69007) were maintained in SFX-insect medium (HyClone, Logan, UT, SH30278.02) at 28˚C. Human HEK 293T cells (ATCC, Manassas, USA, CRL-3216) and SRSF2 protein-depleted SRSF2-MEFs were maintained in DMEM (GIBCO, NY, USA, C11965500BT) supplemented with 10% FBS (BI, Beit-Haemek, Israel, 04-001-1A) or tet-free FBS (Clontech, CA, USA, 631106) under humidified 5% CO2/95% air at 37˚C. SRSF2-MEFs were treated with 10 μg/mL doxycycline (DOX, Sigma, MO, USA, D9891-1G) for 1 day to deplete SRSF2 transcription (DOX+) [48]. We used HEK 293T cells to express the locust SRSF2-V5 fusion protein because this cell line represents a rapid and high-yield protein production system and express more locust SRSF2 than *Drosophila* S2 cells. Lastly, we adopted the SRSF2-MEF cell line to verify the interaction of SRSF2 with *PAHAL* by using an endogenous SRSF2-KO system constructed in the cell line.

### RNA isolation, 5′ and 3′ RACE and qPCR

Nymphal tissues were rapidly dissected and stored in liquid nitrogen for ≤ 2 months before RNA preparation. The total RNA was extracted by using TRIzol reagent (Invitrogen, CA, USA, 15596018) and incubated for 15 min at room temperature (RT) with 15 kunitzK units of RNase-free DNase I (Qiagen, Hilden, Germany, 79254) to remove genomic DNA. The RNA quality, integrity and quantity were detected using NanoDrop 2000c (Thermo, CA, USA) and 1% agarose gel electrophoresis. Analyses using 5′ and 3′ RACE were performed with a SMARTer RACE cDNA amplification kit in accordance with the manufacturer's instructions (Clontech, CA, USA, 634858). The PCR products were purified and cloned into a pGEM-T vector (Promega, WI, USA, A1360) for sequencing. The cDNA was reverse-transcribed with a FastQuant RT Kit (Tiangen, Beijing, China, KR106-02). Briefly, 2 μg of DNase-treated total RNA was mixed with 1 μL of reverse transcriptase mix, 2 μL of FQ-RT primer mix and 2 μL of Fast RT buffer into RNase-free water to obtain a final volume of 20 μL. The mixture was incubated for 30 min at 42˚C. The cDNA was attenuated to 100 μL and stored at −20˚C. Each 10 μL qPCR reaction contained 2 μL of cDNA template, 5 μL of SYBR Green I Master and 0.5 μM of upstream and downstream primers. The qPCR was performed with LightCycler 480 SYBR Green I Master kit (Roche, Mannheim, Germany, 4887352001) on a LightCycler 480 instrument (Roche, Switzerland). Predenaturation was performed at 95˚C for 10 min, followed by

45 cycles of PCR at 95˚C, 58˚C and 68˚C for 20 s. The amplification specificity of the target genes was assessed on the basis of a melting curve. All PCR products were verified through sequencing before qPCR. The housekeeping gene ribosomal protein 49 (*RP49*) was used as the endogenous control in qPCR analysis [67]. The relative expression levels of the specific genes were quantified by using the $2^{-\Delta Ct}$ method, where ΔCt is the Cp value of *RP49* subtracted from that of the gene of interest. Four to eight biological replicates and three technical replicates for every replicate were prepared for each treatment. The outliers were removed from the qPCR standard curves on the basis of the results of Grubbs' test [69]. S3 Table illustrates all primers for RACE and qPCR.

### Sense-specific RT-PCR

The sense-specific PCR was performed as previously described with slight modifications [56]. Briefly, 2 μg of DNase-treated total RNA was mixed with 2 pmol of a reverse primer (primer 13 in Fig 1C) for reverse transcription by using SuperScript IV Reverse Transcriptase (Thermo Fisher; Vilnius, Lithuania, 18090050). The PCR was performed with Taq polymerase (Takara, Tokyo, R001A). An initial denaturation of 94˚C for 3 min was followed by 94˚C for 30 s, 58˚C for 30 s, 72˚C for 40 s for 30 cycles. Next, primers 10 and 12 were used to specifically amplify *PAH* transcript, and primers 9 and 10 were used to specifically amplify the *PAHAL* transcript. The PCR with primers 11 and 12 is expected to yield no product if *PAH* and *PAHAL* are different transcripts. S3 Table present these primers.

### 5′ exonuclease digestion assay

5′-Phosphate-dependent exonuclease digestion was performed with a terminator 5′-phosphate-dependent exonuclease kit in accordance with the manufacturer's instructions (Epicentre, IL, US, TER51020). Briefly, total RNA isolated from locust brains was treated with the exonuclease, which specifically digests RNA species with a 5′-monophosphate end, at 30˚C for 60 min. The 5′-capped mRNA was isolated through phenol extraction and ethanol precipitation, and cDNA was generated using high-capacity RNA-to-cDNA kit (ABI, CA, USA, 4387406). The mRNA fraction of *PAHAL*, *GAPDH* and 18S rRNA was analysed through semi-quantitative PCR. *GAPDH* and 18S rRNA were used as positive and negative controls, respectively.

### Poly(A) tail identification

Poly(A) tails were detected with the Poly(A) tail length assay kit (USB Corporation, Vilnius, Lithuania, 76455). Briefly, a limited number of guanosine and inosine residues were added to the 3′ ends of poly(A)-containing RNAs that were prepared from locust brains. The tailed-RNA was converted to cDNA through reverse transcription by using the newly added G/I tails as the priming sites. *PAHAL U6* and *GAPDH* were amplified from the cDNA with gene-specific primers (S3 Table). The RT-PCR of *PAHAL*, *U6* and *GAPDH* gave a fragment of 242, 71 and 180 bp, respectively.

### *In vitro* translation assay

TNT Quick for the PCR DNA kit (Promega, WI, USA, L5540) was used for the *in vitro* translation assay to assess the coding capability of *PAHAL*. The assay was performed in accordance with the manufacturer's instructions. Sense primers containing T7 promoters were used to amplify the *Renilla* luciferase gene (coding negative control) and the full-length *PAHAL* sequence. The PCR products (1 μg) were mixed with 0.75 μg of transcend biotin–lysyl–tRNA and 1 μL of methionine (1 mM) in 40 μL of TNT T7 Quick Master Mix to a final volume of

50 μL. The mixture was incubated for 1 h at 37˚C. Thereafter, 15 μL of the reaction mix was added into 50 μL of the rehydration/sample lysis buffer (8 M urea, 2 M thiourea, 1% SDS and 0.02% [wt/vol] β-mercaptoethanol) in a tube and mixed by inverting the tube until the mixture became homogeneous. After incubation overnight at RT, 10 μL of the homogenised samples was mixed with 1 μL of 1% bromophenol blue buffer (1% bromophenol blue and 50 mM Tris base) and subjected to polyacrylamide gel (15%) electrophoresis. The samples were then transferred to polyvinylidene difluoride (PVDF) membranes (Millipore, CA, USA, ISEQ00010). The membranes were blocked with 5% (wt/vol) skimmed milk at RT for 1 h. The newly synthesised protein containing biotin was detected by using a chemiluminescent nucleic acid detection module (Pierce, CA, USA, 89880).

### RNA interference and RNA-seq

ds*PAHAL*, ds*PAH* and ds*SRSF2* were used to knock down the *PAHAL*, *PAH* and *SRSF2* expression, respectively. This method has been widely used for specific and efficient knockdown in the locust [40,70–72]. The dsRNAs of *GFP* (ds*GFP*) were used as the negative control. The dsRNAs were synthesised by using the T7 RiboMAX express RNAi system (Promega, WI, USA, P1700). Three *PAHAL*-specific siRNAs (Fig 1B) were designed for siRNA interference. The siRNAs were synthesised by RiboBio (Guangzhou, China) and used as a siRNA pool to interfere with the *PAHAL* expression. The short interfering RNAs of *GFP* (si*GFP*) were used as the negative control. The brains of G locusts were injected with 69 nL of dsRNAs (2 μg/μL) or siRNAs (2 μg/μL) with a glass micropipette tip mounted on a nanoliter injector (World Precision Instruments, FL, USA) under an anatomical lens. The injected locusts were returned to normal rearing conditions and reared for 3 days before their brains were harvested for RNA or protein extraction. Three independent biological replicates were prepared for the RNA-seq analysis of ds*GFP* and ds*PAHAL*. The integrity of total RNA was quantified with an Agilent 2100 Bioanalyser (Agilent). The cDNA libraries were prepared and sequenced in accordance with the manufacturer's protocol of Illumina Nova-seq 6000 (150 PE) at Berry Genomic Corporation, Ltd., Beijing, China. HISAT2 software was used to acquire clean reads from the raw data and map them with the locust genome sequence. The gene-expression level was calculated using the clean reads per kb million mapped reads by using the HTseq tool. Genes with corrected $P < 0.05$ were considered differentially expressed. Differentially expressed genes were defined as corrected $P < 0.05$ and log2|FoldChange|$> 1$. The expression levels were shown by a heat map signal that indicates log2 fold-change values relative to the median expression level within the group. The high and low expression levels relative to the median expression level within the group are represented by yellow and blue signals, respectively. KEGG enrichment was performed by using KOBAS software. Significance was analysed through Fisher's exact test.

### Liquid chromatography–mass spectrometry (LC–MS)

Brains were immediately dissected and stored in liquid nitrogen before being assayed. Ten brains were collected for each of the ten biological replicates. Each sample was homogenised with 100 μL of ice-cold PBS. Thereafter, 10 μL of 1 M perchloric acid was added to 90 μL of the homogenate for protein precipitation. The homogenate was centrifuged at 5200 ×*g* for 30 min at 4˚C. The supernatants were filtered through 0.22 μm filters (Millipore, MA, USA). An equal volume of methanol (Fisher, NJ, USA) was added to the filtered supernatant of each sample for immediate LC–MS assay. The total protein of the remaining homogenate was applied as the loading control and quantified through the BCA assay (Fisher, NJ, USA, 23227).

LC–MS was performed with a rapid resolution liquid chromatography system (ACQUITY UPLC I-Class, Waters, USA). Hydrophilic interaction chromatography separation was

performed with an ACQUITY UPLC™HSS PFP column (100 mm × 2.1 mm, 1.8 μm). The autosampler was set at 10˚C. The mobile phase A of the system was gradient elution with 0.1% formic acid acetonitrile (Pierce, CA, USA; Fisher, NJ, USA), whereas phase B was 0.1% formic acid water. The linear gradient for A was as follows: 10%, 0–3 min; 10%–15%, 3–5 min; 15%–100%, 5–7 min; 100%, 7–8 min; 100%–10%, 8–8.1 min; and 10%, 8.1–11 min. The flow rate was adjusted to 0.2 mL/min, and the injection volume was 10 μL. The total run time for each sample was 11 min.

LC/MS/MS analysis was performed on AB SCIEX Triple Quad 4500 (Applied Biosystems, CA, USA) with an electrospray ionisation source (Turbo Ion spray). Mass spectrometry detection was performed in positive electrospray ionisation mode. The [M + H] of the analyte was selected as the precursor ion. The quantification mode was the multiple-reaction monitoring (MRM) mode using mass transitions (precursor/product ions).

The ESI ion source temperature was 500˚C. The other mass spectrometric parameters were as follows: curtain gas flow: 10 psi; collisionally activated dissociation gas setting: medium; ion spray voltage: 5500 V; and ion gas 1 and 2: 50 psi. Acquisition and processing were performed by using AB SCIEX Analyst 1.6 Software (Applied Biosystems, CA, USA). The collision energies for different MRM pairs were individually optimised with octopamine (m/z 154 > 136, CE 11V), DA (m/z 154 > 137, CE 14 V), serotonin (m/z 177 > 160, CE 17 V), tyramine (m/z 138 > 77, CE 38 V), L-dopa (m/z 198.1 > 152.1, CE 18 V), 5-hydroxytryptophan (m/z 221.1 > 204.1, CE 14 V) and tyrosine (m/z 182.1 > 136.1, CE 18 V).

## Fluorescence in situ hybridization (FISH)

A double FISH experiment was performed as previously described with slight modifications [73]. The RNA probes for *PAHAL* (digoxigenin [DIG]–*PAHAL*) and *PAH* (biotin–*PAH*) were synthesised by using a T7/SP6 RNA polymerase kit (Promega, WI, USA, P2075, P1085) and DIG RNA labelling mixture/biotin RNA labelling mix (Roche, Mannheim, Germany, 11277073910, 11685597910). Nymphal brains were fixed in 4% (wt/vol) paraformaldehyde for 30 min at RT or overnight at 4˚C. The brain tissue was embedded in 5% agarose. Hardened brains were trimmed and dissected into 40 μm slices with a Leica VT1200 S Vibrating Microtome (Leica, Bensheim, Germany). Brain slices were washed twice with PBST (0.5% Triton X-100 in 1× PBS) and then soaked in PBST for 10 min at RT to permeabilise the brain tissue. Brains were digested with proteinase K (160 μg/mL; Invitrogen, CA, USA, AM2548) at 37˚C for 20 min and then washed thrice with PBST. Prehybridisation was performed by using a prehybridisation buffer (Wuhan Boster, Wuhan, China, AR0152) at 37˚C for 30 min. Brain slices were hybridised with DIG–*PAHAL* and biotin–*PAH* probes (5 ng/μL) at 37˚C overnight and then blocked at 4˚C in 2% BSA (2% BSA in 0.2× SSC) for 20 min. Next, brain slices were incubated in anti-DIG alkaline phosphatase-conjugated antibody (1:500; Roche, Mannheim, Germany, 11093274910) and streptavidin–HRP (1:100) for 1 h at RT and then washed thrice with PBS. The fluorescent signal of DIG for *PAHAL* or that of biotin for *PAH* was detected with an HNPP Fluorescent Detection Set (Roche, Mannheim, Germany, 11758888001) or TSA Fluorescein System (Perkin-Elmer, MA, USA, NEL701A001KT). Images were captured under a LSM 710 confocal fluorescence microscope (Zeiss, Oberkochen, Germany) at 10× and 40× magnifications. S3 Table lists the primers used for the probe synthesis of *PAHAL* and *PAH*.

## Isolation and crowding of locusts

The locusts were isolated by introducing typical G nymphs into metal boxes and individually rearing them under standard conditions. The locusts were crowded by introducing 10 labelled

S nymphs and 20 G nymphs into an optic Perspex box (10 cm × 10 cm × 10 cm). Adequate fresh food was provided. After 0, 4 or 8 h of treatment, the locust brains were dissected and immediately frozen in liquid nitrogen for RNA preparation. All insects were sampled at the same time of the day to avoid the effects of circadian rhythm on the locust phenotypes. Equal numbers of male and female insects were sampled for each biological replicate.

## Behavioural assays

Behavioural assays were performed as previously described [40,42]. A rectangular Perspex arena (40 cm × 30 cm × 10 cm) was used in the assay. One of the separated side chambers (7.5 cm × 30 cm × 10 cm) contained 30 G locusts as a stimulus group. Another chamber with the same dimensions was left empty. A locust was released into the centre of the arena and monitored for 300 s. Individual behavioural data were automatically recorded and analysed with the EthoVision video tracking system (v.3.1.16, Noldus Inc., Wageningen, Netherlands). The three behavioural variables were as follows: total distance moved (TDM) and total duration of movement (TDMV), which represent motor activity levels, and attraction index (AI, i.e., total duration in stimulus area minus total duration in opposite area), which represents the attraction or repulsion to the stimulus group. The behavioural phase state of each locust was assessed by applying a single probabilistic metric of gregariousness $P_{greg}$, $P_{greg} = e^{\eta}/(1 + e^{\eta})$, where $\eta = -2.11 + 0.005 \times AI + 0.012 \times TDM + 0.015 \times TDMV$ [36,42]. $P_{greg}$ indicates the probability of a locust regarded as G. $P_{greg} = 1$ indicates fully G behaviour, whereas $P_{greg} = 0$ means fully S behaviour.

## Nuclear fractionation

Nuclear and cytoplasmic fractionation was performed as previously described [73]. Twenty nymphal brains were harvested and homogenised in a cold lysis buffer [1× PBS containing 0.2% IGEPAL CA-630 (Sigma, MO, USA, I8896-50ml) and 1× proteinase inhibitor (Pierce MA, USA, 88266) and RNase inhibitor (Promega, WI, USA, N2111S)] for nuclear fractionation. The homogenate was then centrifuged at 30 ×$g$ for 2 min at 4˚C. The supernatants were transferred to a fresh tube and centrifuged at 425 ×$g$ for 15 min at 4˚C to obtain the nuclear pellet. The cytoplasmic fraction in the supernatant was centrifuged at 2000 ×$g$ for 10 min at 4˚C to remove residual nuclei. The nuclear pellet and cytoplasmic supernatant were maintained at −80˚C prior to RNA extraction.

## Reporter and expression plasmid construction, luciferase assay and antibodies

Luciferase assays were performed to verify whether *PAHAL* regulates *PAH* in *cis*. Plasmids containing different promoter regions of *PAH* and their deletions were constructed into pGL4.10 (Promega, WI, USA, E665A) by using the KpnI and XhoI restriction sites. Promoter regions were amplified from locust genomic DNA. A series of ESEs with mutations in the *PAH* 5′-UTR (containing −554/+89 fragment) were inserted into pGL4.10 obtained from Pole-Polar Biotechnology Co., Ltd., Beijing, China.

The full-length *PAHAL* sequence, the *PAHAL* sequence in the reverse orientation (reverse *PAHAL*), the *PAHAL* sequence with deletions, the *SRSF2* ORF and a 3 kb *lacz* ORF (as the negative control) were cloned into the pcDNA3.1 (+) vector (Invitrogen, CA, USA, V79020) and/ or pAc5.10/V5-His A vector (Invitrogen, CA, USA, V4110-20) by using KpnI and XhoI. These vectors were transfected with *Drosophila* S2 cells, HEK 293T cells and SRSF2-MEFs.

Cells were plated on 48-well plates and transfected by using Lipofectamine 3000 in accordance with the manufacturer's instructions (Invitrogen, CA, USA, L3000015). Reporter

plasmids (10 ng) with 200 ng of the expression plasmid or negative control vector were co-transfected with 5 ng of the internal control plasmid pRL-TK/pGL4.73 (Promega, WI, USA, E2241/ E691A). Luciferase activity was measured with Dual-Luciferase Reporter Assay System (Promega, WI, USA, E1960) at 30 h after incubation.

The polyclonal antibodies for PAH and monoclonal SRSF2 antibody were produced by immunising mice with the prokaryotic expression peptide (Beijing Protein Innovation, Beijing, China). The tubulin (rabbit) and GAPDH (rabbit) antibodies [74] were provided by Dr. Yun-Dan Wang.

## RNA pulldown and Western blot analysis

Twenty brains in one biological duplicate were lysed to acquire total protein for RNA pull-down by using 200 μL of tissue lysis buffer [200 μL of T-PER tissue protein extraction reagent (Pierce, CA, USA, 78510) and 2 μL of Halt Protease Inhibitor Cocktail, EDTA-free (Pierce, CA, USA, 87785)]. Such an undertaking was carried out to identify the interactive proteins with *PAHAL in vivo*. RNA pulldown was performed with a magnetic RNA–protein pull-down kit in accordance with the manufacturer's instructions (Pierce, CA, USA, 20164). The RNA probes for full-length *PAHAL* and their deletions (biotin–*PAHAL*) were synthesised with a T7 RNA polymerase kit. The RNA-associated proteins were separated on 15% SDS-PAGE gel and then subjected to protein silver staining. Single silver-stained bands that were present in *PAHAL* pulldown but absent from antisense *PAHAL* pulldown were excised and then bleached by using a destaining buffer (50% acetonitrile and 25 mM NaHCO3). Disulphide bonds of protein samples were disrupted by adding 10 mM DTT for 1 h at 56˚C and 55 mM Iodoacetamide for 45 min at RT. Protein was hydrolysed by incubation in trypsin buffer (62.5 ng/μL trypsin and 25 mM NaHCO3) at 37˚C overnight. The protein samples were subjected to mass spectrometry (Beijing Protein Innovation, Beijing, China). MASCOT software (Multiple Alignment System developed by iCOT) [75] was used to identify and quantity proteins from the mass spectrometric data. The protein functions were annotated by the protein scores > 85, which is a criterion of the significant high expression level of proteins ($P < 0.05$) in this single silver-stained band.

RNA pulldown and western blot analysis were performed on SRSF2-MEFs ($2 \times 10^7$) that co-transfected with pcDNA3.1/V5-His/*SRSF2* ORF and a series of *PAHAL*-pcDNA3.1+ deletions. This task was carried out to identify the specific sites for SRSF2 interaction in the *PAHAL* sequence.

The total proteins for Western blot analysis were first extracted by using TRIzol reagent. The proteins were dissolved in rehydration/sample lysis buffer (*in vitro* translation assay) to a concentration of 10 μg/μL overnight at RT and then mixed with 1 μL of 1% bromophenol blue buffer. The proteins were then separated through SDS-PAGE on 10% NuPAGER Bis–Tris gel (Invitrogen, CA, USA, NP0315BOX) by using 1× NuPAGE MOPS SDS running buffer (Invitrogen, CA, USA, NP0050) in accordance with the manufacturer's instructions. The separated proteins were transferred to PVDF membranes by using 1× transfer buffer and blocked with 5% (wt/vol) skimmed milk for 1 h at RT. The membranes were incubated in a blocking buffer at 4˚C overnight with a primary antibody in the following concentration: anti-PAH (mouse), 1:500; anti-SRSF2 (mouse), 1:500; anti-SmD1 (rabbit), 1:2,000; anti-GAPDH (rabbit), 1:5,000; and anti-Tubulin (rabbit), 1:5,000. The secondary antibody (1:5,000; Easybio, Beijing, China, BE0101-100, BE0102-100) in the blocking buffer was incubated for 1 h at RT. The immunological blot was detected with SuperSignal West Femto Substrate Trial Kit (Pierce, CA, USA, 34094).

## RNA immunoprecipitation (RIP) assay

The *SRSF2* ORF was first cloned in frame with the V5 epitope of the pcDNA3.1/V5-His vector for the construction of pcDNA3.1/V5-His/*SRSF2* ORF to test the binding of SRSF2 with *PAHAL in vitro*. HEK 293T cells were then co-transfected with this vector and pcDNA3.1 (+)/*PAHAL* for the following RIP assay. After 3 days, $2 \times 10^7$ of HEK 293T cells were harvested for the RIP experiment.

The RIP assay was performed with Magna RIP Quad RNA-Binding Protein Immunoprecipitation Kit (Millipore, CA, USA, 17–704). The cell pellet was homogenised in ice-cold RIP lysis buffer containing 1× proteinase inhibitor and RNase inhibitor and stored at −80˚C overnight. Magnetic beads were pre-incubated with 5 μg of V5 antibody (Invitrogen, CA, USA, R96025) or normal mouse IgG (Millipore, CA, USA, CS200621) for 30 min at RT with rotation. The supernatants of the lysate from the centrifugation were co-incubated with the bead–antibody complex overnight at 4˚C with rotation. Thereafter, 10 μL of the supernatants was stored as the input. The RNA in the immunoprecipitates and input was extracted by using TRIzol reagent. Reverse transcription was performed with a high-capacity RNA-to-cDNA kit. The target gene expression was analysed through qPCR.

The binding of SRSF2 with *PAHAL in vivo* was tested by performing the RIP assay on brain tissues. A Dynabeads Protein G immunoprecipitation kit (Thermo Fisher, CA, USA, 10007D) was used. S4 Table shows the antibody epitopes of PAH and SRSF2. Fifty brains in one biological replicate were lysed with 700 μL of T-PER Tissue Protein Extraction Reagent containing RNasin Plus RNase Inhibitor (Promega, WI, USA, N2611) and EDTA-free 1× Halt Protease Inhibitor Cocktail. After the lysates were centrifuged at 10,000 ×$g$ for 10 min, the supernatants were collected and subjected to RIP by following the protocol included with the kit. A total of 100 μL of Dynabeads was pre-incubated with 10 μg of antibody for 10 min at RT with rotation. The supernatants (10 μL) of the lysate were stored as the input. A total of 300 μL of supernatants were co-incubated with the bead-specific antibody complex with rotation for 10 min at RT. The RNA and proteins were extracted using a TRIzol reagent before quantitative measurement.

## Bioinformatics and statistical analysis

Transcriptome data subjected to poly(A) lncRNA screening were obtained from a previous publication [76]. Locust lncRNA sequences were predicted and identified through an integrative method as previously described [44,57]. Sequence conservation of *PAHAL* was analysed by blasting in the genome assembly of all species in flybase (www.flybase.org). The sequence motifs were WNNNNSNNAGCCC (W = A/T, S = G/C) for the NLS [47] and WSSNGYY (W = A/T, S = G/C Y = C/T) for the SRSF2-responsive ESE [49,77]. Data from the tissue expression and truncated luciferase assay were analysed through ANOVA, then by post-hoc Tukey's *b*-test for multiple comparisons. Independent sample Student's *t*-tests were performed for comparing differences in gene expression and other values between treatments. The frequency data of behavioural features, namely, $P_{greg}$ and AI, were analysed through Mann–Whitney *U* test. Two-sided *P*-values were provided. Data are presented as the mean ± SEM unless stated otherwise. All statistical data were analysed by using SPSS 21.0 (SPSS Inc., IL, USA). The locust genome data are available at the following website: http://www.locustmine. org. All sequences for *PAHAL*, *PAH* and *SRSF2* have been deposited in GenBank under accession numbers KX962170, KX951493 and KX951494, respectively. The RNA-seq data have been uploaded to NCBI with the accession number PRJNA522953. Numerical data that underlies graphs or summary statistics and sample image data in support of all reported results have been uploaded to Harvard Dataverse Network with the websit https://dataverse.harvard.edu/

## Supporting information

**S1 Table. The FPKM of differently expressed gene.**
(XLSX)

**S2 Table. The proteins from *PAHAL* pulldown identified by mass spectrometry.**
(DOCX)

**S3 Table. Sequences of all primers used in the study.**
(DOCX)

**S4 Table. The antibody epitopes of PAH and SRSF2.**< /SI_Caption>
(DOCX)

**S1 Fig. Full-length sequence of the *PAH* locus.** The red bold text represents the exon
sequence of *PAH*. The bold text with underscored characters represent the full-length
sequence of the *PAHAL*.
(PDF)

**S2 Fig. Localisation of *PAHAL* and *PAH* in FISH.** The nuclei of cell bodies in locust brain
neurons are stained by Hochest 33342 (blue signal). A purple signal indicates that the nuclear
location of *PAHAL*. 10× (left images) and 63× (right images) magnification.
(TIF)

**S3 Fig. Nuclear localisation signal (NLS) identified in *PAHAL*.** The bold text represents the
NLS sequence "WNNNNSNNAGCCC" (W = A/T, S = G/C). The underscored characters represent the conserved nucleic acid sequence.
(TIF)

**S4 Fig. Gene expression of proteins from *PAHAL* pulldown during locust isolation and
crowding.** (A) mRNA levels of 10 proteins in nymphal brains during isolation. The 10 proteins
(excluding SRSF2) identified from *PAHAL* pulldown pools. Fig 6C shows SRSF2. The mRNA
level was determined through qPCR. The crowding responses of three candidate genes that
showed changes in the expression in response to locust isolation were measured. (B) mRNA
levels of the three candidate genes during nymphal crowding. The *SmD1* gene (LOCMI17236)
with prominent density response was chosen for further tests. Eight biological replicates of
eight brains were prepared for each treatment. Means labelled with the same letter within each
treatment are insignificantly different. (C) Western blot analysis for the detection of the non-
specific association of SmD1 with *PAHAL*. Tubulin was used as a control (n = 4). Four biological replicates of eight brains were measured in each treatment. (D) RIP *in vivo* revealed no
association between SmD1 and *PAHAL*. Nymphal brains were harvested for RIP with SmD1
antibody or control IgG. Five biological replicates of 50 brains were prepared for each treatment. Error bars represent ± SEM.
(TIF)

**S5 Fig. SRSF2 binding with *PAHAL*.** (A) Western blot analysis of locust SRSF2-V5 fusion
protein. The protein was expressed in HEK 293T by transfecting with pcDNA3.1/V5-His/
*SRSF2* and detected using V5 antibody or the locust SRSF2-specific antibody. (B) SRSF2 was
detected in lysed brains immunoprecipitated with the locust SRSF2-specific antibody or IgG
control. Data were normalised to IgG. Five biological replicates of 50 brains were measured.
Student's *t*-test: *$P < 0.05$. Error bars indicate ± SEM. (C) Distribution of SRSF2 binding

elements (ESEs) in the promoter-proximal region of *PAH*. The three predicted ESEs, namely, ESE1, ESE2 and ESE3, in the 5′-UTR of *PAH* gene are highlighted. "+1" represents the TSS of *PAH*. "+89" represents the end of 5′-UTR of *PAH*. The green characters represent the mutant sequence of ESE. Student's *t*-test: $^*P < 0.05$, $^{**}P < 0.01$; n.s., insignificant.
(TIF)

## Acknowledgments

We are grateful to Xiang-Dong Fu for providing SRSF2-MEFs, Yun-Dan Wang for providing tubulin and GAPDH antibodies, Yi Ding and Weihua Wang for assisting in LC-MS, and Fangqing Zhao for the useful discussion.

## Author Contributions

**Conceptualization:** Bing Chen, Le Kang.

**Data curation:** Xia Zhang, Bing Chen.

**Formal analysis:** Xia Zhang, Bing Chen, Le Kang.

**Funding acquisition:** Bing Chen, Le Kang.

**Investigation:** Xia Zhang, Ya'nan Xu.

**Methodology:** Xia Zhang, Bing Chen.

**Project administration:** Bing Chen, Le Kang.

**Resources:** Le Kang.

**Supervision:** Bing Chen, Le Kang.

**Validation:** Xia Zhang, Bing Chen, Le Kang.

**Visualization:** Xia Zhang, Bing Chen.

**Writing – original draft:** Xia Zhang, Bing Chen.

**Writing – review & editing:** Bing Chen, Le Kang.

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
