## [Decision Letter · Decision Letter 0]

21 Feb 2020

Dear Dr Kang,

Thank you very much for submitting your Research Article entitled 'Long noncoding RNA PAHAL modulates locust behavioural plasticity through the feedback regulation of dopamine biosynthesis' to PLOS Genetics. Your manuscript was fully evaluated at the editorial level and by independent peer reviewers. The reviewers appreciated the attention to an important topic but identified some aspects of the manuscript that should be improved, and which may require a re-review of the revised manuscript.

We therefore ask you to modify the manuscript according to the review recommendations before we can again consider your manuscript for acceptance. Your revisions should address the specific points made by each reviewer.

[LINK]

Yours sincerely,

John Ewer

Associate Editor

PLOS Genetics

Gregory P. Copenhaver

Editor-in-Chief

PLOS Genetics

Reviewer's Responses to Questions

**Comments to the Authors:**

Reviewer #1: Summary:

Zhang et al. have presented an interesting and thorough mechanistic investigation into the role of phenylalanine hydroxylase (PAH) in brain biogenic amine metabolism and transcriptional regulatory networks. The aggregation switching behavior in locusts has been studied from a variety of perspectives and the authors build upon a significant number of studies that specifically indicate the role of dopamine and biogenic amines in this phenotypic transition. I found this paper in general to be methodologically strong and integrative, and the experiments across domains (targeted / high-throughput gene expression analysis, metabolic measurements, PCR, behavioral analysis) were linked clearly. The transcriptional activation experiments were very nice, and demonstrated that the authors were committed to exploring the exact functional of the PAHAL lncRNA – they passed a high level of evidence to show that the PAHAL transcriptional activation was relevant in multiple contexts (rather than e.g. simply identifying a putative lncRNA transcript from RNA-seq data and publishing a hypothesis paper). My comments are all in service of clarifying the experiments, results, and interpretations of the authors.

Main comments.

I am somewhat confused by Figure 2B. Usually, lncRNA regulate loci in trans, and are transcribed from loci that are distal to lncRNA targets (see comment 2 though). However Figure 2B looks like PAHAL has a transcriptional start site within the 4th intron of PAH, is transcribed in the same direction as PAH (e.g. at Line 102: “ The transcription direction of this gene [PAHAL] is the same as that of the canonical PAH (Fig 1B)”), and includes the 4th intron + 5th exon + part of the 5th intron (the red section). Is this the case? If so, then PAHAL’s sequence would be anti-sense to PAH (e.g. the PAHAL lncRNA would anneal to the sense strand of PAH so it would have sequence homology to the anti-sense PAH). Then I am confused by the primer design in Figure 1C -- the common reverse primer #11 is located within PAH exon 6, which is not within PAHAL as visualized, but apparently this means that Exon 6 is within PAH (as discussed in text, e.g. At Line 301). At Line 101 it is written: “The PAHAL sequence covers the sixth exon and part of the two introns flanking the exon in the PAH gene (Fig 1B and S1 Fig).”. First off, this not what I see in Figure 1B (Line 923: “The red shading indicates the location of PAHAL in PAH locus)”, where the red PAHAL transcript appears to be from the 4th intron to the middle of the 5th intron (and thus doesn’t include the 6th exon as suggested at Line 101). Then the Supplemental Figure S1 appears to be a screenshot of a fasta file of the sequence of the 3' end of PAHAL transcript, highlighting the NLS sequence, but not providing any information about which introns/exons of PAH are included within PAHAL (this Figure S1 should be more clearly informative, as it is used to support the idea at Line 101 “The PAHAL sequence covers the sixth exon and part of the two introns flanking the exon in the 102 PAH gene (Fig 1B and S1 Fig)”, which I am not seeing). So honestly I cannot tell exactly where the PAHAL locus/transcript begins and ends. For example in Figure 1C, the PCR with 10+9 gave no product, when this is what spans Exon 5 to Intron 5, I’d expect this to form a product with the unspliced PAH or PAHAL transcript. The caption mentions that the locus diagrams are not to scale – I think for a complex focal loci in this paper, a scale figure would be more appropriate.

Related to my comment 1: I appreciated the citations in Lines 295-299 about cis-regulatory lncRNA, but are there any citations to which % of lncRNA act in this manner? Otherwise these two cited examples could be the two rare exceptions. Or it could be discussed why we do not / cannot know which fraction of lncRNA may act in this way.

This is a comment about the RNAi experiments and interpretation. Line 116: “We knocked down PAHAL through dsRNA and siRNA interferences to test the regulatory relationship between PAHAL and PAH in the locust brain.”. I think that the unique relationship of PAH-PAHAL makes the dsRNA/siRNA experiments inconclusive, or at least not completely explained. Here is my contention: since dsRNA targeting the PAHAL locus also (by definition) targets the PAH locus, the dsRNA is directly knocking down PAH mRNA (as well as PAHAL) – in fact both transcripts are reduced an almost-identical amount, and follow the same regulatory time course in later experiments. The authors interpret their knockdown experiment solely in terms of “the dsRNA knocks down PAHAL mRNA, and the reduction of PAHAL mRNA reduces PAH”. However it is more parsimonious to suggest that the dsRNA corresponding to the PAH/PAHAL locus knocks down transcription of that entire locus, thus not necessarily acting via the transcriptional loop hypothesized. The piece I might be missing here is if the siRNA causes a “PAHAL-specific knockdown”, as suggested at Line 120. But again, any siRNA corresponding to PAHAL would correspond to an intron/exon of PAH and thus be likely to knock down PAH transcription directly – in other words, the entire logic of this section/experiment is that the dsRNA/siRNA experiments only knock down PAHAL (hence interpreting PAH decreases as mechanistically downstream of PAHAL), but because PAHAL is inside PAH, any RNAi-type experiments would be more simply interpreted as a transcriptional silencing of the PAH locus directly. I believe the correct control (or at least an experimental group to include in these experiments) for these dsRNA experiments would be a dsRNA/siRNA targeting a region in PAH but not PAHAL (e.g. in 1st exon of PAH). This would hypothetically reduce PAH without reducing PAHAL, and if this phenocopies the “PAHAL knockdown” presented here, would be strong support for the author’s interpretation. Also as a related note, just because a RNAi knockdown reduces a metabolite X, this does not mean that that the knocked down gene is a rate-limiting step in the synthesis of X.

I think that the sentence at Line 293-294, that the positive transcriptional influence of PAHAL was verified in “locust, fruit fly, and mice in vivo and in virto” is extremely strong. I think that this framing could be introduced earlier – that the effect of any candidate genes identified through high-throughput approaches would be verified through in vivo and in vitro experiments in multiple species.

The SRSF2 experiments were very nice, and good support for the article’s main hypotheses.

A large body of research suggests that the rate limiting step of Dopamine biosynthesis is Tyrosine Hydroxylase (TH), and TH is used commonly as the defining marker of dopaminergic neurons (e.g. to provide just one citation: ncbi.nlm.nih.gov/pmc/articles/PMC3065393/ ). In the Figure 1a of this manuscript, it is clearly laid out that TH and DDC are both required after PAH for DA synthesis. At Line 72 the authors write that PAH “is a rate-limiting enzyme in the synthesis of DA and other bioamines [34,35]”, but citations 34 & 35 do not specifically support the idea that PAH is indeed rate-limiting for DA or any amine other than Tyrosine. In the Discussion from Lines 259-266 there are good citations relating to the specific role of PAH in locust, these citations could be relevant in the intro as well. Otherwise, I believe this statement that links PAH directly as rate-limiting to DA synthesis is inaccurate (it is TH, not PAH). In general the manuscript is oriented towards the specific role of PAH in DA synthesis, and this ends up neglecting non-PAH players in DA metabolism & PAH’s role in other processes.

In general, whenever a P value is presented, I would appreciate full statistical information (e.g. Line 162 and 168 mention the statistical test used, add information on the N, and test statistic). In many other cases, I was unclear which statistical test was being performed, and how large the sample sizes were, without combining back through Methods/Materials/etc.

Minor comments:

Line 20: As is, this sentence sounds like PAHAL is what "is known for extreme behavioral plasticity", not the locusts. To clarify the sentence, something like "...behavioural adjustment in migratory locusts, a species with extreme behavioral plasticity". Also the latin species name could be included in the abstract, for search reasons and clarity.

Line 22, 42, elsewhere: Repeatedly this lncRNA is described as being “is sense to PAH gene”. This use of the word “sense” may be obscure for those who are unfamiliar with the sense/anti-sense terminology. Perhaps in an early mention, something like “PAHAL has sequence homology to the sense (coding) strand of the PAH locus” or “PAHAL ” (or whatever accurate statement would clarify my uncertainty in the next sentences).

Line 77: Would be helpful to add an estimate of how long "rapidly shift their behaviours" is referring to. E.g. does "rapidly" mean 1 hour, 1 day, 1 week.

Line 128-131: This sentence makes it sound like the phenylethylamine (tyrosine/dopa/dopamine/octopamine) and tryptamine (5HTP/serotonin) biogenic amines are all downstream of PAH. Rather, as pointed out a few lines later, there are two separate branches (one downstream of phenyalanine/PAH, and one downstream of tryptophan). The authors then claim at Line 135 that “These results indicate that 136 PAHAL only affects the DA synthesis in the catecholamine metabolic pathway”, which is incorrect – the knockdown of PAH (whether directly or via PAHAL transcriptional loop) resulted in changes to tyrosine plus some phenylethylamines (l-dopa/dopamine but not octopamine/tyramine), and also changes to 5HTP levels (why do the authors think that this tryptamine would be decreased by PAH knockdown when it is not linked to phenylethylamine metabolism?).So while it is true that “PAHAL only affects the DA synthesis in the catecholamine metabolic pathway”, this section would benefit from an explanation or interpretation of the unexpected changes to 5HTP on the non-catecholamine branch.

Line 145-146: “The result showed that PAH and PAHAL mRNA were mainly expressed in cell bodies” – as opposed to what? The cell body is where all transcription and gene expression occurs (even if RNA is transported or localized elsewhere, it is still “expressed” from the nucleus/body). So I don’t know what the point of this sentence is, as all genes are expressed in the cell body (maybe the authors mean to say that the transcripts are localized only in the cell bodies).

In Figure 2E I would prefer if the Y-axes in Figure 2 were not truncated – this makes the differences between the groups appear to be much larger than they actually are.

Figure 3B -- I do not understand why the "Rlative mRNA" (typo, should be "Relative") starts at 0.015, instead of 1. What is this mRNA being relativized to?

If the raw RNA-seq reads have been uploaded to a public database (e.g. Short Read Archive), this could be included (or perhaps it already here and I missed it).

Grammatical suggestions:

Line 42: “mediated the transcription activation” –> “transcriptional”

Line 57: “exhibits the spatiotemporal specific expression patterns” can delete “the”.

Line 60: “regulate the gene expression” can delete “the”

Line 66: "is"  "are"

Reviewer #2: Zhang et al., describe the neural expression a locust long non-coding RNA (lncRNA), PAHAL, and provide good evidence that it regulates expression of the gene encoding phenylalanine hydroxylase (PAH) by binding to the SRSF2 splicing factor and recruiting it to the PAH promoter. Evidence is presented that this regulatory mechanism controls dopamine (DA) amounts in the locust brain. The authors show that PAHAL is expressed in the brain (and in the nucleus) and that both it and PAH are elevated when locusts are in crowded conditions and decreased when the insects are isolated. This suggests a role for the lncRNA in the regulation of social behavior. Further, they show that RNAi-mediated reductions in PAHAL result in a transition from group to solitary behavior, supporting the idea that it can regulate such behavior. Additional studies indicate that PAHAL binds to a short segment of the proximal promoter region of PAH, providing a mechanism for regulation of the gene. Using immunoprecipitation assays and brain tissues, the authors show that PAHAL is bound to SRSF2, in vivo, and that SRSF2 amounts increase with locust crowding and decrease in solitary locusts, similar to what is observed for PAHAL and PAH. They use cell-based assays to show that SRSF2 enhances expression of PAH and that it interacts with PAHAL to regulate gene expression. Finally, the authors identify the sequences within PAHAL that bind SRSF2 for its recruitment to PAH as well as exonic-splicing enhancer (ESE) sites within PAH that mediate regulation by SRSF2. All of these results lead to a satisfying model in which PAHAL recruits SRSF2 to the PAH gene to regulate its expression, dopamine signaling and behavior.

I have no major issues with the manuscript, but the following should be addressed prior to publication:

-The Discussion section should be shortened considerably; it is far too long.

-What was the reason for thinking PAHAL regulated PAH. Was it just the proximity of PAHAL to PAH or was there another reason for believing that PAHAL might be regulatory? The authors may wish to state this explicitly.

-I was initially confused by the PAHAL knockdown experiment because it seemed like the dsRNA would target both PAH and PAHAL. I then realized that the dsRNA was targeted to an intron of PAH, so it ought to be specific for PAHAL. The authors should add a sentence in the appropriate section making this clear so that other readers will not be confused.

-p. 6, Please indicate somewhere in this first paragraph that primer sequences for making dsRNA and siRNAs can be found in supplemental materials.

-p 8, it might be good here to remind the reader about G and S populations at the beginning of this section, even though they were described in the Intro. The average reader won't be as familiar with locust behavior as the authors.

-p 8, the images in Fig. 3C are difficult to interpret unless one happens to be a locust neurobiologist. Which cell bodies are shown? Are the RNAs expressed everywhere in the brain or is there a more specific expression pattern? Do the images represent the whole brain or part of the brain? Also, it looks like PAHAL is predominantly nuclear whereas PAH is cytoplasmic so I'm not sure if colocalization is the right word to use. I would say they are in the same cells with one being nuclear and the other cytoplasmic. Similarly, in Fig. 5B, what brain region is shown?

-p. 10, top, does siRNA or dsRNA for PAH and PAHAL have the same effects on behavior. Perhaps this is known from a previous publication. If so, please mention it.

-p. 11, bottom, please state clearly that SRSF2 was tagged with V5. The wording here is a little unclear.

-p 13, there is reference to Fig 6C which I think should be 7C.

-Fig. 6 is missing the legend for panel F.

-It's almost impossible to read the names of the genes shown in Fig 2C. Please create a larger figure for publication.

-In some places PAHAL is referred to as mRNA. It’s not really a messenger RNA as it doesn’t encode a protein. It seems better to refer to it as RNA, not mRNA.

Reviewer #3: The authors investigate the regulation of the gene phenylalanine hydroxylase (PAH) in the locust, which plays a role in dopamine production in the brain that, in turn, regulates behavioral phase changes (solitary vs. gregarious). A model is proposed wherein a portion of the unspliced PAH transcript is separately transcribed as a sense long noncoding RNA (PAHAL) that positively regulates PAH transcription by recruiting a splicing factor protein that interacts with the PAH promoter. The regulatory potential of long noncoding RNAs is an exciting area of research. The authors present an impressive array of experimental data, including gene expression assays, RNA interference, RNA immunoprecipitation, mouse cell knockouts, and luciferase assays of PAHAL (and interacting factors) effects on transcription. The results of these experiments provide insights into PAH regulation important to dopamine-mediated locust phase transitions, but I have some concerns about the robustness of the described role of the long noncoding RNA, PAHAL, that is proposed to mediate neurotransmitter metabolism – and behavioral plasticity by orchestrating a local transcriptional loop.

Could the PAHAL dsRNA used to target PAHAL also target PAH pre-mRNA (since both are in the nucleus) to directly decrease PAH expression?

Did the nuclear fractionation experiment differentiate PAHAL from PAH pre-mRNA?

For the luciferase assays, is it possible that the PAHAL enhancer effect is simply due to a genomic enhancer element residing in an intron? In other words, could it be that the effect on transcription is not typically RNA-mediated but is rather through DNA looping interactions with the promoter? Can the authors comment on whether/how their experiments preclude this possibility?

Could the RNA pulldown with SRSF2 be capturing pre-mRNA in the nucleus that is being targeted by the spliceosome (including SRSF2, which is associated with nascent RNA and the spliceosome)?

Line 81: I did not see PAH mentioned in the abstract or in a word search of ref 41. Can the link between microRNA-133 and PAH in this sentence be described more clearly? Was PAH knocked down in this prior study?

The senior author has argued that other single factors like a microRNA are responsible for modulating dopamine and behavioral plasticity in locusts in other publications. Can the authors take an integrative view in their discussion to describe how the findings in the present manuscript fit in with their prior findings about dopamine regulation in the locusts?

Overall, I think it is an overstatement to conclude that “PAHAL modulates behavioral aggregation” as stated in the abstract and that “PAHAL mediates PAH transcriptional activation.” Some of my hesitance to accept these statements arises because it seems possible that PAHAL is produced only in response to PAH activation to further enhance transcription (what activates PAHAL?). I would like to see the authors respond to potential sources of uncertainty outlined above in my comments and more explicitly lay out the logic that counteracts such concerns or otherwise introduce more uncertainty to the interpretation of their findings where appropriate.

**Have all data underlying the figures and results presented in the manuscript been provided?**

Reviewer #1: Yes

Reviewer #2: Yes

Reviewer #3: None

PLOS authors have the option to publish the peer review history of their article (what does this mean?). If published, this will include your full peer review and any attached files.

Reviewer #1: No

Reviewer #2: No

Reviewer #3: No

---

## [Decision Letter · Decision Letter 1]

2 Apr 2020

Dear Dr Kang,

Thank you very much for submitting your Research Article entitled 'Long noncoding RNA PAHAL modulates locust behavioural plasticity through the feedback regulation of dopamine biosynthesis' to PLOS Genetics. Your revised manuscript was fully evaluated at the editorial level and again by independent peer reviewers. The reviewers appreciated your thorough responses to their comments. There is now only one issue pending, which was raised by Reviewer #2; we ask that you address this issue before we can consider your manuscript for acceptance.

This next version will not be sent out again for review; instead the newly revised version will be evaluated at the editorial level. To assist us in this job, please indicate in your cover letter how you have responded to this comment, and in your manuscript please highlight the text you have added/modified. 

In addition we ask that you upload a Striking Image with a corresponding caption to accompany your manuscript if one is available (either a new image or an existing one from within your manuscript). If this image is judged to be suitable, it may be featured on our website. Images should ideally be high resolution, eye-catching, single panel square images. For examples, please browse our archive. If your image is from someone other than yourself, please ensure that the artist has read and agreed to the terms and conditions of the Creative Commons Attribution License. Note: we cannot publish copyrighted images.

[LINK]

Yours sincerely,

John Ewer

Associate Editor

PLOS Genetics

Gregory P. Copenhaver

Editor-in-Chief

PLOS Genetics

Reviewer's Responses to Questions

**Comments to the Authors:**

Reviewer #1: Summary:

Zhang et al. have revised their paper which investigated phenylalanine hydroxylase (PAH) in the migratory locust (Locusta migratoria).

In general the responses in blue text to my comments were excellent. In several of the responses to my comments, and the responses to Reviewer #3, the authors demonstrate a very reasoned understanding of RNA and neurotransmitter biology.

I appreciated the addition of the RNAi control experiments, the edits to the Figures, and the addition of accessible datasets (which I have not downloaded and investigated in detail, but trust that the authors have uploaded all info into an authorized database).

Other than that, I am satisfied with the revisions and have nothing more specific to add. With the key changes to the cited literature about lncRNA, PAH functionality, and PAHAL experiments, the manuscript is significantly improved.

Reviewer #2: Regarding my comments about Fig. 3, I understand that locust cell bodies are in a cortex surrounding the neuropil; this is similar to Drosophila. Rather, I was referring to the localization of PAH and PAHAL mRNAs within neuronal cell bodies. PAH is cytoplasmic whereas PAHAL is nuclear so strictly speaking they do not co-localize (i.e., they are in different cell compartments). A better way to describe their localization is to say something like: "They are both localized to neurons, but as expected PAH is cytoplasmic whereas PAHAL is nuclear".

And, the authors still need to indicate whether PAH and PAHAL are expressed in all neurons or just a subset (e.g., DA neurons). Fig. 3 makes it look like PAH and PAHAL have broad expression patterns rather than being localized only in a subset of locust neurons. Please clarify.

And, further, where are the aminergic neurons localized in the locust brain? This must be known. A comparison of the observed PAH and PAHAL expression patterns with the known location of aminergic neurons would be informative. I don't expect the authors to verify the location of aminergic neurons, just to point out their general location in the locust brain in comparison to the PAH and PAHAL expression patterns.

Reviewer #3: The authors’ responses to my questions and concerns were thoughtful and thorough. Well done.

**Have all data underlying the figures and results presented in the manuscript been provided?**

Reviewer #1: Yes

Reviewer #2: Yes

Reviewer #3: Yes

PLOS authors have the option to publish the peer review history of their article (what does this mean?). If published, this will include your full peer review and any attached files.

Reviewer #1: No

Reviewer #2: No

Reviewer #3: No

---

## [Editor Report · Decision Letter 2]

9 Apr 2020

Dear Dr Kang,

We appreciate the care you have taken in addressing the most recent comments from reviewers and are pleased to inform you that your manuscript entitled "Long noncoding RNA PAHAL modulates locust behavioural plasticity through the feedback regulation of dopamine biosynthesis" has been editorially accepted for publication in PLOS Genetics. Congratulations!

Yours sincerely,

John Ewer

Associate Editor

PLOS Genetics

Gregory P. Copenhaver

Editor-in-Chief

PLOS Genetics

Comments from the reviewers (if applicable):

**Data Deposition**

http://datadryad.org/submit?journalID=pgenetics&manu=PGENETICS-D-20-00072R2

**Press Queries**

---

## [Editor Report · Acceptance letter]

20 Apr 2020

PGENETICS-D-20-00072R2 

Long noncoding RNA *PAHAL* modulates locust behavioural plasticity through the feedback regulation of dopamine biosynthesis 

Dear Dr Kang, 

We are pleased to inform you that your manuscript entitled "Long noncoding RNA *PAHAL* modulates locust behavioural plasticity through the feedback regulation of dopamine biosynthesis" has been formally accepted for publication in PLOS Genetics! Your manuscript is now with our production department and you will be notified of the publication date in due course.

With kind regards,

Laura Mallard

PLOS Genetics

On behalf of:
